# DNA methyltransferase 3A controls intestinal epithelial barrier function and regeneration in the colon

Antonella Fazio[1,8], Dora Bordoni[1,8], Jan W. P. Kuiper[1], Saskia Weber-Stiehl[1], Stephanie T. Stengel[1], Philipp Arnold[2], David Ellinghaus [1], Go Ito[1,3], Florian Tran [1,4], Berith Messner[1], Anna Henning[1], Joana P. Bernardes[1], Robert Häsler[1,5], Anne Luzius[1], Simon Imm[1], Finn Hinrichsen[1], Andre Franke [1], Samuel Huber [6], Susanna Nikolaus[4], Konrad Aden [1,4], Stefan Schreiber[4], Felix Sommer [1], Gioacchino Natoli[7], Neha Mishra[1,9] & Philip Rosenstiel [1,9] ✉

Genetic variants in the DNA methyltransferase 3 A (DNMT3A) locus have been associated with inflammatory bowel disease (IBD). DNMT3A is part of the epigenetic machinery physiologically involved in DNA methylation. We show that DNMT3A plays a critical role in maintaining intestinal homeostasis and gut barrier function. DNMT3A expression is downregulated in intestinal epithelial cells from IBD patients and upon tumor necrosis factor treatment in murine intestinal organoids. Ablation of DNMT3A in Caco-2 cells results in global DNA hypomethylation, which is linked to impaired regenerative capacity, transepithelial resistance and intercellular junction formation. Genetic deletion of *Dnmt3a* in intestinal epithelial cells (*Dnmt3a*^*ΔIEC*^) in mice confirms the phenotype of an altered epithelial ultrastructure with shortened apical-junctional complexes, reduced Goblet cell numbers and increased intestinal permeability in the colon in vivo. *Dnmt3a*^*ΔIEC*^ mice suffer from increased susceptibility to experimental colitis, characterized by reduced epithelial regeneration. These data demonstrate a critical role for DNMT3A in orchestrating intestinal epithelial homeostasis and response to tissue damage and suggest an involvement of impaired epithelial DNMT3A function in the etiology of IBD.

Intestinal epithelial cells (IECs) have evolved to build a complex physiological barrier that separates the intestinal lumen from the underlying mucosal tissue. This barrier protects against invading pathobionts, while at the same time allowing controlled access to antigens and a peaceful co-existence of the human host with the commensal resident microbiota[1]. Several functional components of the epithelium contribute to this unique role, among them the secretion of a mucus layer and life-long regeneration of the different epithelial cell types from a stem cell compartment, allowing for quick repair of barrier defects[2].

Impaired epithelial barrier function is regarded as a key feature of patients with inflammatory bowel disease (IBD)[3]. IBD is a multifactorial disorder known to be strongly influenced by genetic susceptibility and environmental factors[4], although the interplay between these factors is still poorly understood. Thus, epigenetic mechanisms as long-term determinants of gene expression are a plausible link between genetics and environment, which may instigate the manifestation and/or progression of IBD. Indeed, disease-specific DNA methylation signatures of peripheral blood cells and intestinal biopsies were identified in adult IBD patients[5–7]. In children diagnosed with IBD, genome-wide DNA

methylation profiling of purified intestinal epithelial cells clearly distinguishes between healthy individuals and the IBD group[8].

DNA methyltransferases (DNMTs) constitute the enzyme family responsible for the establishment and maintenance of DNA methylation patterns. Whereas DNMT1 is the key maintenance methyltransferase, DNMT3A and DNMT3B represent the main de novo methyltransferases, as they can bind unmethylated DNA and establish novel DNA methylation marks not only during early development but also in differentiated cells in a signal-dependent manner[9]. Both enzyme activities cooperate to establish and maintain cellular methylation patterns. Although DNMT3A and DNMT3B are highly homologous, their redundancy seems to be limited, and their unique biological function, e.g., specificity for methylation patterns in distinct cell types, is still elusive[9]. Genetic deficiency induces embryonic lethality in Dnmt1 and Dnmt3b-deleted mice and postnatal lethality in Dnmt3a-knockout mice at ~4 weeks of age[10]. Interestingly, a genome-wide association study has shown that noncoding polymorphisms in the de novo methyltransferase enzyme DNMT3A locus are associated with a higher risk for IBD, and for Crohn´s disease (CD) in particular[11]. The exact role of DNMT3A in the context of intestinal inflammation has not been investigated so far. In this work, we analyze expression patterns of DNMT3A in patients with IBD, perform loss of function experiments using Caco-2 cell line and generate a conditional *Dnmt3a*[ΔIEC] mouse line. We show that DNMT3A is reduced in epithelial cells of IBD patients and demonstrate that DNMT3A-driven epigenetic regulation of gene expression is required for forming proper epithelial junction zones and barrier function. Loss of functional DNMT3A in IECs leads to increased susceptibility to experimental colitis, which suggests a mechanism for how genetic variants in the DNMT3A locus could precipitate into the manifestation of Crohn´s disease.

## Results

### DNMT3A expression is downregulated in intestinal epithelial cells of CD patients
To investigate the role of DNMT3A in CD, we first determined whether IBD leads to transcriptional dysregulation of DNMT3A in humans. We performed qRT-PCR analysis of whole biopsy samples obtained from 60 patients (n = 30 CD; n = 30 UC;) and 30 healthy individuals[12]. We found that *DNMT3A* mRNA levels, but not *DNMT3B* nor *DNMT1*, were significantly reduced in both inflamed and non inflamed samples from Crohn´s disease (CD), but also from ulcerative colitis patients, compared to healthy individuals (Fig. 1a, Supplementary Table 1). We next focused on the regulation of DNMT3A in intestinal epithelial cells. Therefore, we generated human intestinal organoids from CD patients and healthy individuals. We observed a significant downregulation of *DNMT3A* transcript levels, together with a reduced expression of the proliferative marker *CCND1* and increased expression of a pro-inflammatory marker *CXCL10* in epithelial organoids from endoscopically inflamed and non inflamed biopsies from CD patients compared to healthy controls at early passages (Fig. 1b). We showed that DNMT3A is also reduced at the protein level in organoid cultures derived from colonic as well as small intestinal epithelial cells from CD patients compared to healthy controls. This effect was even more pronounced in organoids derived from inflamed tissue (Fig. 1c, Supplementary Fig. 1a and Supplementary Table 2). Of note, DNMT3A1 represents the main isoform expressed in intestinal epithelial cells.

To understand potential inflammatory triggers responsible for the observed downregulation of DNMT3A, we next employed murine intestinal organoid cultures to circumvent genetic and environmental variability that human organoid might carry over[13]. Among the prototypic pro-inflammatory stimuli, i.e., tumor necrosis factor-α (TNF), but not interferon-γ (IFN) and bacterial lipopolysaccharide (LPS) led to a significant downregulation of *Dnmt3a*. This effect was not observed for the other de novo methyltransferase Dnmt3b (Supplementary Fig. 1b), while we observed increased gene expression of pro-inflammatory markers such as *Cxcl1*, *Cxcl10,* and *Tnf* (Supplementary Fig. 1c) upon TNF and LPS treatment. The results indicate that pro-inflammatory stimuli, such as TNF, may inhibit DNMT3A expression in intestinal epithelial cells, although the sensitivity of IECs to specific signals may differ between mice and humans.

Finally, we tested for which genes genetic risk at the known CD risk locus 2p23.3 (including DNMT3A)[14,15] is mediated by genetically regulated expression (Supplementary Fig. 1d). We performed gene expression imputation and transcriptome-wide association analysis (TWAS) using GWAS summary statistics of 5956 CD cases and 14,927 controls as well as 6968 UC cases and 20,464 controls and whole-blood RNA-seq and genome-wide genotype reference data from 922 individuals. DNMT3A expression was significantly associated with genetic risk for IBD and particularly for CD ($p$DNMT3A = $2.93 \times 10^{-6}$). It was the only gene at 2p23.3 to achieve transcriptome-wide significance (Supplementary Data 1; $p$Bonferroni = $0.05/11475 = 4.36 \times 10^{-6}$ for 11,475 imputable genes), with the negative Z score indicating that the genetic risk variant for CD at this locus is associated with reduced gene expression of DNMT3A. The results indicate not only the complex involvement of environmental but also genetic factors, which may act in synergy to reduce DNMT3A expression in Crohn´s disease.

### Deletion of DNMT3A induces a complex transcriptional dysregulation and hypomethylation in intestinal epithelial cells
We next aimed to understand the functional consequences of a loss-of-function of DNMT3A in IECs. To this end, DNMT3A was genomically deleted in Caco-2 cells by CRISPR-Cas9 genome editing (hereafter named ΔDNMT3A). Additionally, the two known DNMT3A isoforms (DNMT3A1 and DNMT3A2) were individually reintroduced in the ΔDNMT3A cell line to identify genes specifically regulated by the respective DNMT3A isoforms (Fig. 2a). Complete ablation of DNMT3A expression in the ΔDNMT3A cell line, and subsequent re-expression of DNMT3A1 or DNMT3A2 isoforms was confirmed by immunoblot detection (Supplementary Fig. 2a). The subcellular localization of DNMT3A was assessed by immunofluorescence staining and confirmed that DNMT3A was mainly confined to the nuclei[16], while it could not be detected in the ΔDNMT3A cells. When re-expressed, DNMT3A1 and DNMT3A2 both were also localized in the nuclei (Supplementary Fig. 2b). RNA-sequencing analysis identified over 4000 differentially expressed genes (DEGs) which were significantly dysregulated upon deletion of DNMT3A. The dysregulated expression of 1820 DEGs was restored upon reintroduction of the two DNMT3A isoforms (Supplementary Fig. 2c, Supplementary Fig. 2e, Supplementary Data 2, Supplementary Data 3). Interestingly, we observed several transcripts that were only rescued by one of the isoforms, i.e., by either DNMT3A1 or DNMT3A2, suggesting an isoform-dependent regulation (Fig. 2b). Parallel to RNA-sequencing, we performed DNA methylation profiling on WT, ΔDNMT3A, DNMT3A1, and DNMT3A2 cells using a genome-wide based bead array interrogating >850,000 methylation sites. Analysis of differentially methylated regions (DMRs) demonstrated a global hypomethylation in ΔDNMT3A cells (Fig. 3a, Supplementary Data 4). We also observed a minority of significantly hypermethylated sites indicating an influence of DNMT3A deletion on the activity of other DNMTs, while their mRNA levels were unchanged in the RNA-seq datasets. In line with the transcriptome profile, re-expression of DNMT3A isoforms rescued the methylation profile of 69,184 out of 81,750 differentially methylated positions (DMPs) between WT and ΔDNMT3A cells. In addition, DMRs were more frequently found to overlap with proximal enhancers, suggesting the possible functional impact of DNA methylation changes on gene expression in specific genomic regions (Fig. 3b).

To identify mRNA expression changes associated with *cis*-linked methylation changes, we screened all DEGs, which were rescued by

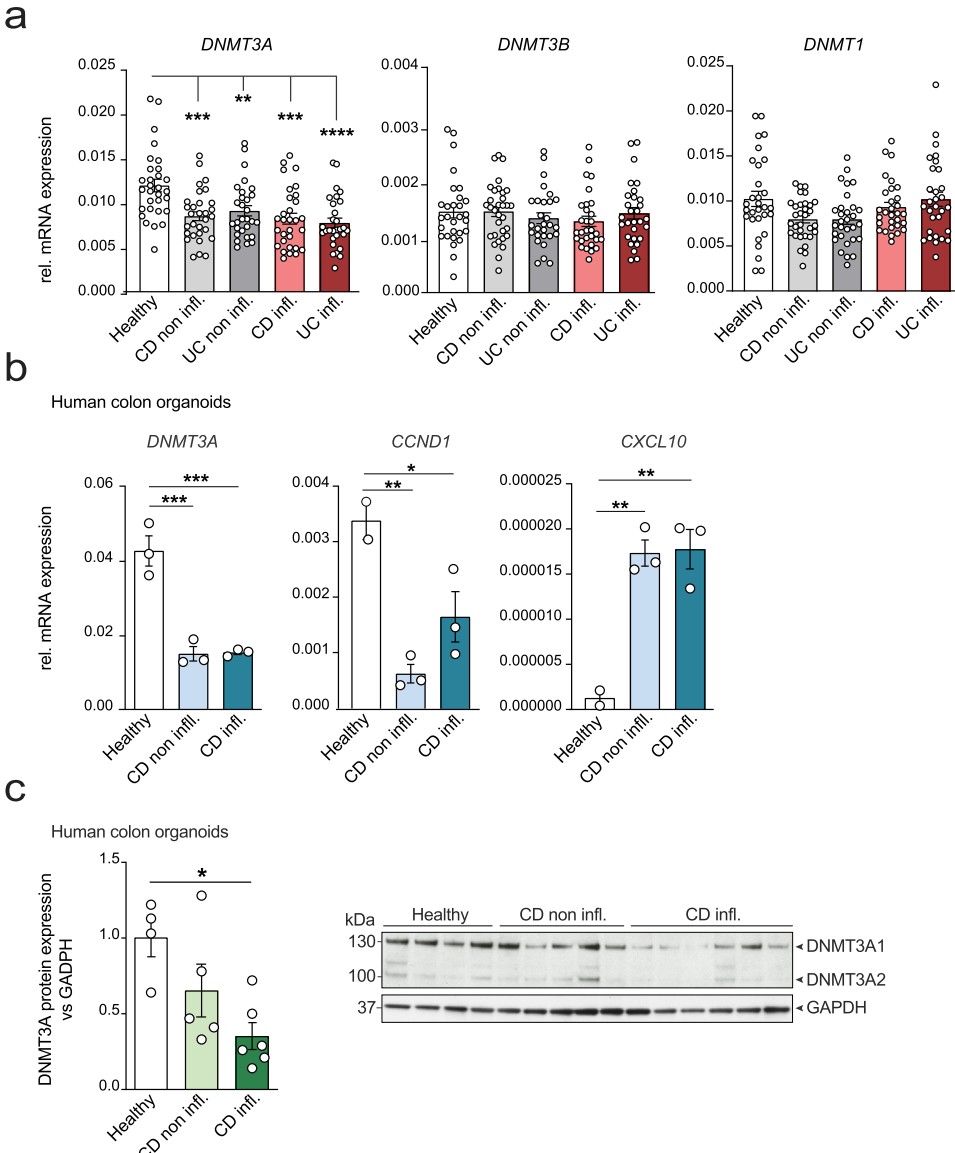

**Fig. 1 | DNMT3A expression is downregulated in IBD patients. a** Relative mRNA expression of *DNMT3A*, *DNMT3B*, and *DNMT1* in biopsies from healthy controls (*n* = 30), CD non inflamed (non infl.) (*n* = 30), UC noninflamed (non infl.) (*n* = 30), CD inflamed (inf.) (*n* = 30), and UC inflamed (inf.) (*n* = 30). Beta-actin was used as housekeeping gene. **b** Gene expression analysis of *DNMT3A*, *CCND1*, and *CXCL10* in human colonic organoids derived from healthy controls (*n* = 3), CD non infl. (*n* = 3), and CD inf. (*n* = 3) patients. Beta-actin was used as housekeeping gene.

**c** Immunoblot and quantification of DNMT3A protein expression in human colonic organoid lysates derived from healthy controls (*n* = 4), CD non infl. (*n* = 5), and CD inf. (*n* = 6) patients. GAPDH protein expression was use for normalization. The values represent mean ± SEM. Statistical analysis was performed using one-way ANOVA together with Tukey post hoc test. *$p < 0.05$, **$p < 0.01$, ***$p < 0.001$, ****$p < 0.0001$.

both DNMT3A isoforms, for DMPs within a 5 kb window up- and downstream of their respective transcription start site (TSS) and calculated the correlation between the expression of the gene and the methylation intensity of each corresponding DMP[17]. Among a total of 1082 rescued DEGs with *cis*-linked methylation changes, expression changes in 386 genes were negatively correlated with the methylation changes in at least one corresponding DMP, whereas 287 genes showed an inverse pattern. Examples of DMP-DEG pairs showing a canonical, inverted relationship between gene expression and DNA methylation (DNAm) include IFNGR1 (interferon gamma receptor 1) and S100A4 (S100 calcium binding protein A4) (Supplementary Fig. 2d, Supplementary Data 5). In line with other studies, non-canonical DNAm-DEG pairs point to more complex *cis*-linked regulatory effects of differential methylation[7]. To identify biological processes which are specifically regulated by DNMT3A in IECs, gene ontology analysis was performed on the set of DEGs that were rescued

by both isoforms. From the set of genes that were upregulated in ΔDNMT3A cells, but corrected by re-expression of either isoform, an enrichment analysis identified biological processes such as type I interferon signaling pathway, response to TNF, and intestinal absorption, among others. Biological processes enriched in the respective downregulated genes, which could be rescued by re-expression, included Wnt signaling, cell–cell communication, and epithelial cell morphogenesis (Fig. 3c, Supplementary Data 6).

**Transcriptome and DNA methylome profiles of intestinal epithelial cells from mice carrying a tissue-specific deletion in DNMT3A**

To understand the role of DNMT3A in IECs in the physiological context, we generated an IEC-specific deletion model of *Dnmt3a* in C57/BL6 mice using a conditional Dnmt3a allele with the exons 13–17 flanked by loxP sites crossed with Villin-Cre mice (termed *Dnmt3a*$^{\Delta IEC}$

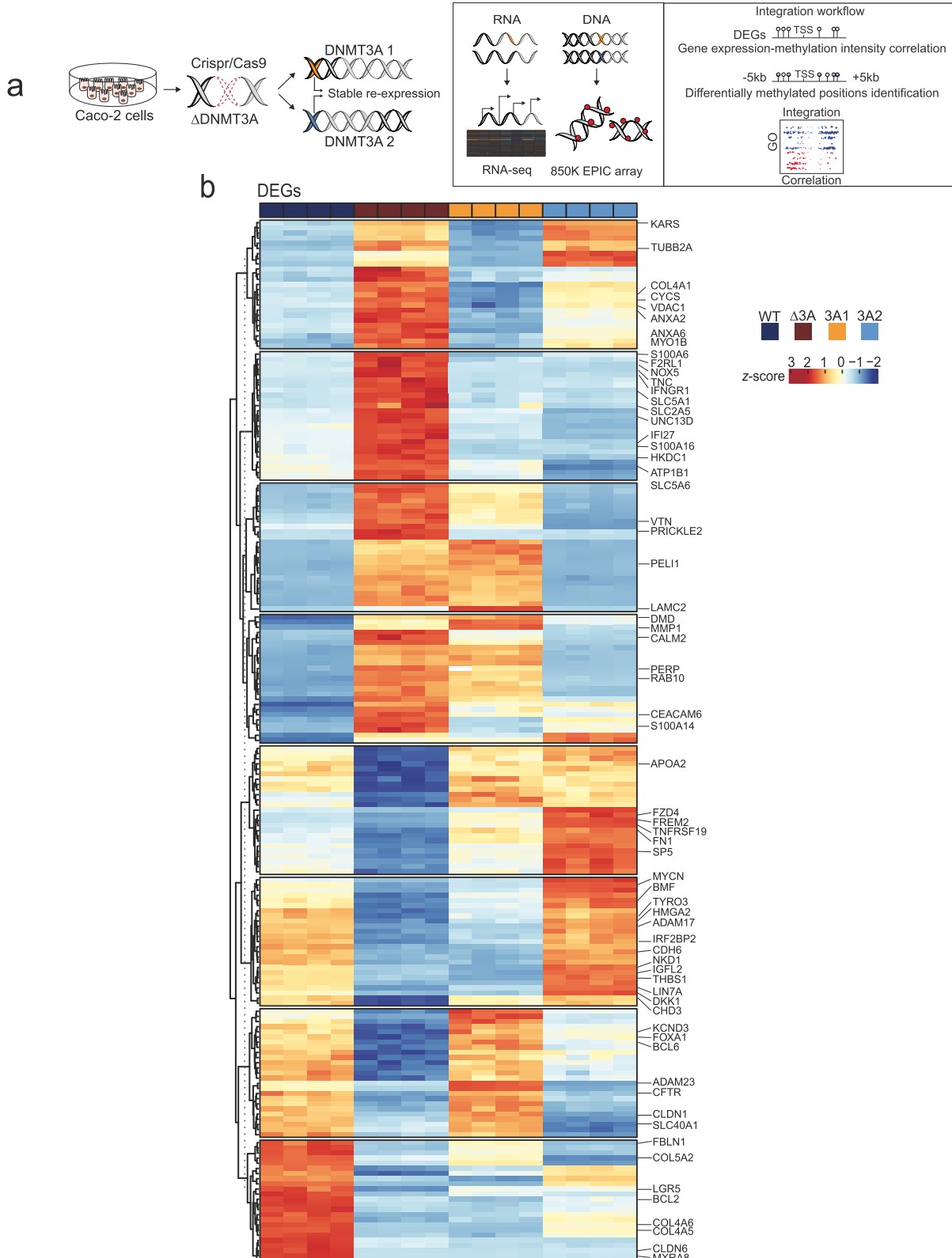

**Fig. 2 | Deletion of DNMT3A results in complex mRNA expression changes in Caco-2 cells. a** Schematic representation of cell lines generation and data analysis workflow. Graphical elements modified from ref. 17. **b** Heatmaps of differentially expressed genes (DEGs) of WT, ΔDNMT3A, DNMT3A1, and DNMT3A2 cell lines ($n = 4$). Scaled gene expression/methylation intensity across all samples are plotted.

hereafter) (Supplementary Fig. 3a–d). In contrast to the lethal phenotype of the full-body knockout animals[10], loss of Dnmt3a in IECs did not impair the development of the mice. Dnmt3a$^{\Delta IEC}$ mice were born at the expected Mendelian ratios and did not show any significant

differences either in body weight, or gross morphology (Supplementary Fig. 3e), confirming the results of a previous study[18]. We further characterized the immune cell composition in the colon and mesenteric lymph nodes, and we did not observe any difference in the

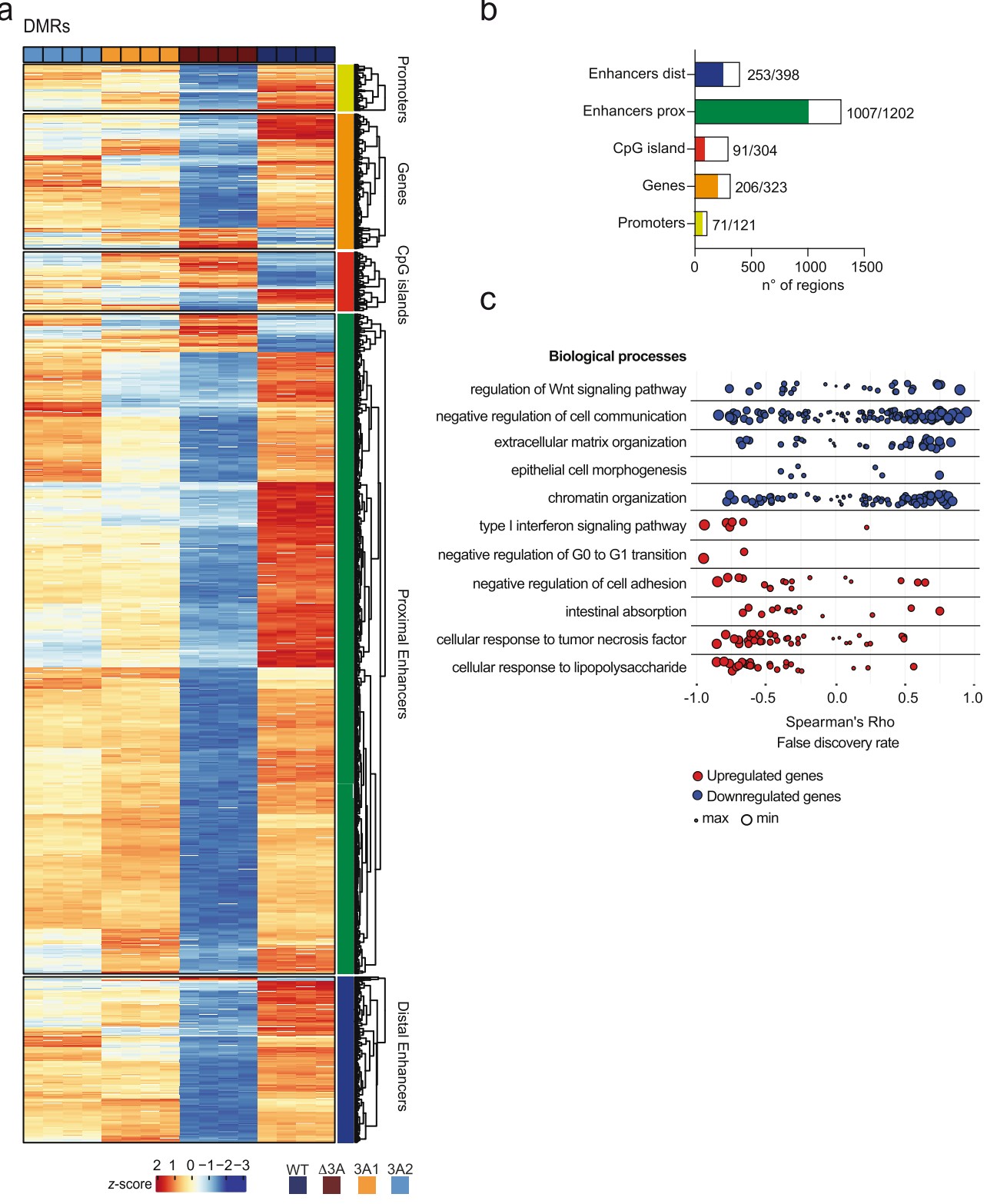

**Fig. 3 | DNA methylation changes in DNMT3A-depleted Caco-2 cells. a** Heatmaps of differentially methylated regions (DMRs) of WT, ΔDNMT3A, DNMT3A1, and DNMT3A2 cell lines ($n$ = 4). Scaled gene expression/methylation intensity across all samples are plotted. **b** Bar plot indicating number of regions differentially methylated in genes rescued by DNMT3A1 and DNMT3A2 isoforms. **c** Correlation between gene expression and DNA methylation of up- (red) and downregulated (blue) rescued genes classified by the GO terms to which the genes belong. Top selected GO terms enriched in the rescued genes are shown. Each point represents a gene, and the size of the points is proportional to statistical significance (FDR) of the correlation with larger points being more significant.

adaptive or innate immune cells frequencies (Supplementary Fig. 3f). We performed DNA methylation and gene expression profiling by genome-wide based bead array interrogating >285,000 methylation sites and RNA-sequencing, respectively, in purified colonic epithelial cells from 12-week old *Dnmt3a^fl/fl* and *Dnmt3a^ΔIEC* mice (Fig. 4a). Dnmt3a deletion caused a global decrease of methylation, across genomic regions with 596 differentially methylated gene bodies and 1023 differentially methylated promoter regions (Fig. 4b, Supplementary Data 7). Transcriptome analysis revealed differential expression of a total of 352 genes, with 237 genes upregulated and 115 genes downregulated in the IECs from *Dnmt3a^ΔIEC* mice (Fig. 4c, Supplementary Data 8). Gene ontology analysis on the differentially expressed genes revealed highly significant dysregulation of genes and processes involved in Wnt signaling pathway, wound healing, and acute inflammatory responses (Fig. 5a, Supplementary Data 9).

Comparing the two models, human Caco-2 cells and purified murine IECs, we checked which of the in vitro DNAm-linked rescued genes are also associated with *cis*-linked DNA methylation changes in their orthologous gene loci in vivo. The intersection identified a set of 446 genes (Fig. 5b, Supplementary Data 5), with approximately half of these genes (221 genes) also showing the same direction of correlation between DEG and DMP. Examples conserved between the mouse and human DNMT3A-deficient state comprise RGS14 (Regulator of G-protein signaling 14) and IFITM3 (Interferon-induced transmembrane protein 3), which showed a canonically increased expression with reduced methylation in the promoter region, whereas COX6B1 (Cytochrome C oxidase subunit 6B1) represents a non-canonical example where higher mRNA levels are linked to increased methylation in the promoter region (Fig. 5c). DNAm-linked DEGs in vivo that were downregulated in *Dnmt3a^ΔIEC* mice were related to cell adhesion and proliferation while the ones upregulated in *Dnmt3a^ΔIEC* mice were mainly involved in inflammatory response (Supplementary Fig. 3g, Supplementary Data 10), consistent with the in vitro results. Key biological processes of intestinal epithelial homeostasis and response to tissue damage, such as wound healing and cell morphogenesis, were enriched in DEGs shared between the in vivo and in vitro models (Supplementary Fig. 3h, Supplementary Data 11 and 12).

## Ablation of DNMT3A alters intestinal ultrastructure and Goblet cell numbers, resulting in decreased intestinal barrier function

We thus next studied morphological and functional changes of IECs upon DNMT3A deletion in greater detail. Caco-2 cells were cultivated in a 3D matrix resulting in the formation of a spheroid-like structure characterized by an inner lumen enclosed by a single layer of polarized cells[19]. While wild-type spheroids were able to form the expected spheroid-like structure, ΔDNMT3A spheroids were not able to form a proper lumen and displayed an altered structural organization compared to the WT control. The spheroid area (μm²) was measured, and ΔDNMT3A spheroids were around 50% smaller than the WT (Fig. 6a). To assess wound healing properties in vitro, WT and ΔDNMT3A Caco-2 cells were employed in an in vitro wound healing assay[20]. Loss of DNMT3A significantly delayed defect closure by 20% compared to the WT control (Fig. 6b). As Wnt signaling is involved in epithelial proliferation and migratory behavior at the wound edges upon injury[21], we next assessed the transcriptional activity of β-catenin using the TOP/FOP luciferase reporter assay in WT and ΔDNMT3A Caco-2 cells. We could show that the deletion of DNMT3A caused a significantly reduced reporter gene expression, indicating reduced autochthonous Wnt signaling activity (Supplementary Fig. 4a).

To further elucidate a potential ultrastructural defect of the cells, we analyzed the apical-junctional complex, which includes tight junctions, adherens junctions, and desmosomes. Differentiated ΔDNMT3A monolayers presented with increased intercellular distance and shortened apical-junctional complexes, when compared to WT cells (Fig. 6c) in transmission electron microscopy (TEM). We also assessed

cell monolayer integrity by measuring the trans-epithelial electrical resistance (TEER). In comparison to WT controls, the ΔDNMT3A monolayer showed a significantly reduced TEER (Fig. 6d). In line with the in vitro model, TEM analysis of small intestine tissue from *Dnmt3a^ΔIEC* mice showed a widened intercellular space and a shortened apical-junctional complex compared to *Dnmt3a^fl/fl* control (Fig. 6e). When intestinal organoids were seeded into a transwell system as a continuous monolayer, the TEER value was also reduced in the *Dnmt3a^ΔIEC* organoids (Fig. 6f).

We next assessed paracellular permeability ex vivo and in vivo. Murine colonic organoids derived from *Dnmt3a^ΔIEC* mice and *Dnmt3a^fl/fl* littermates were treated with the fluorescent dye 4 kDa fluorescein isothiocyanate-dextran (FITC-4D) for 24 h. Fluorescence intensity in the lumen of the organoids was measured and used as an indicator of paracellular permeability. *Dnmt3a^ΔIEC* organoids were more permeable to the fluorescent dye compared to the *Dnmt3a^fl/fl* control (Fig. 6g). We validated these findings also in Caco-2 monolayers, using the fluorescence dye Lucifer Yellow (LY) as paracellular permeability marker. ΔDNMT3A monolayers showed an increased permeability, indicated by higher LY levels in the lower chamber compared to the WT control (Supplementary Fig. 4b). We measured gut permeability in vivo via administration of 4 kDa FITC dextran to *Dnmt3a^ΔIEC* and *Dnmt3a^fl/fl* mice. After 1 h, serum was isolated, and FITC dextran fluorescence was measured. *Dnmt3a^ΔIEC* mice had a higher intestinal permeability compared to *Dnmt3a^fl/fl* littermate control (Fig. 6h).

We assessed the number of secretory Goblet cells in the colon and found a subtle, but significantly reduced Goblet cell count (identified by PAS staining) in *Dnmt3a^ΔIEC* mice compared to their floxed littermate controls (Supplementary Fig. 4c). Since the formation of the protective mucus layer is the prime function of Goblet cells[22], we measured the distance between the epithelium and the luminal bacteria as an indicator of mucus thickness in Carnoy-fixed samples by FISH using a eubacterial probe. In line with the reduced Goblet cell numbers, *Dnmt3a^ΔIEC* mice show a reduced mucus thickness (Supplementary Fig. 4d), compared to *Dnmt3a^fl/fl* mice. As we could not unequivocally identify tissue-bound bacteria in the FISH staining (e.g., intracellular bacteria in epithelial cells or macrophages), we next performed 16 S rDNA qPCR analysis and found a significant increase of bacterial DNA in the mesenteric lymph nodes of *Dnmt3a^ΔIEC* mice compared to WT littermates (Supplementary Fig. 4e). Even if this does not directly translate into spontaneous inflammation, the results indicate that lack of Dnmt3a leads to defects of intestinal barrier function already at a steady state.

## Deletion of intestinal epithelial Dnmt3a results in increased susceptibility to inflammation

To understand the interplay of DNMT3A deficiency and pro-inflammatory signals of immune cells, which are known to impair the barrier function of epithelial cells, we first used an in vitro co-culture model using WT and ΔDNMT3A cells together with THP-1 derived macrophages[23]. To induce an inflammation-like response, differentiated THP-1 cells were pre-stimulated with LPS and IFN-γ. Co-culture of Caco-2 monolayers with activated THP-1 cells induced disruption of the barrier, as indicated by a decreased TEER value. The barrier integrity of the WT co-culture was fully re-established over time, while only incomplete recovery was observed in the ΔDNMT3A co-culture (Supplementary Fig. 4f).

We next studied the role of DNMT3A in IECs during inflammation in mice in vivo. Thus, we first exposed *Dnmt3a^fl/fl* and *Dnmt3a^ΔIEC* mice to chronic DSS-induced colitis[24]. Challenged *Dnmt3a^ΔIEC* mice presented with higher body weight loss (Fig. 7a) and heightened inflammatory disease activity index (DAI score) compared to littermate *Dnmt3a^fl/fl* controls (Fig. 7b). Reduced mRNA levels of the intestinal Lgr5+ stem cell marker *Olfm4* and the proliferative marker *Ccnd1* were detected in inflamed colon tissue of *Dnmt3a^ΔIEC* mice compared with

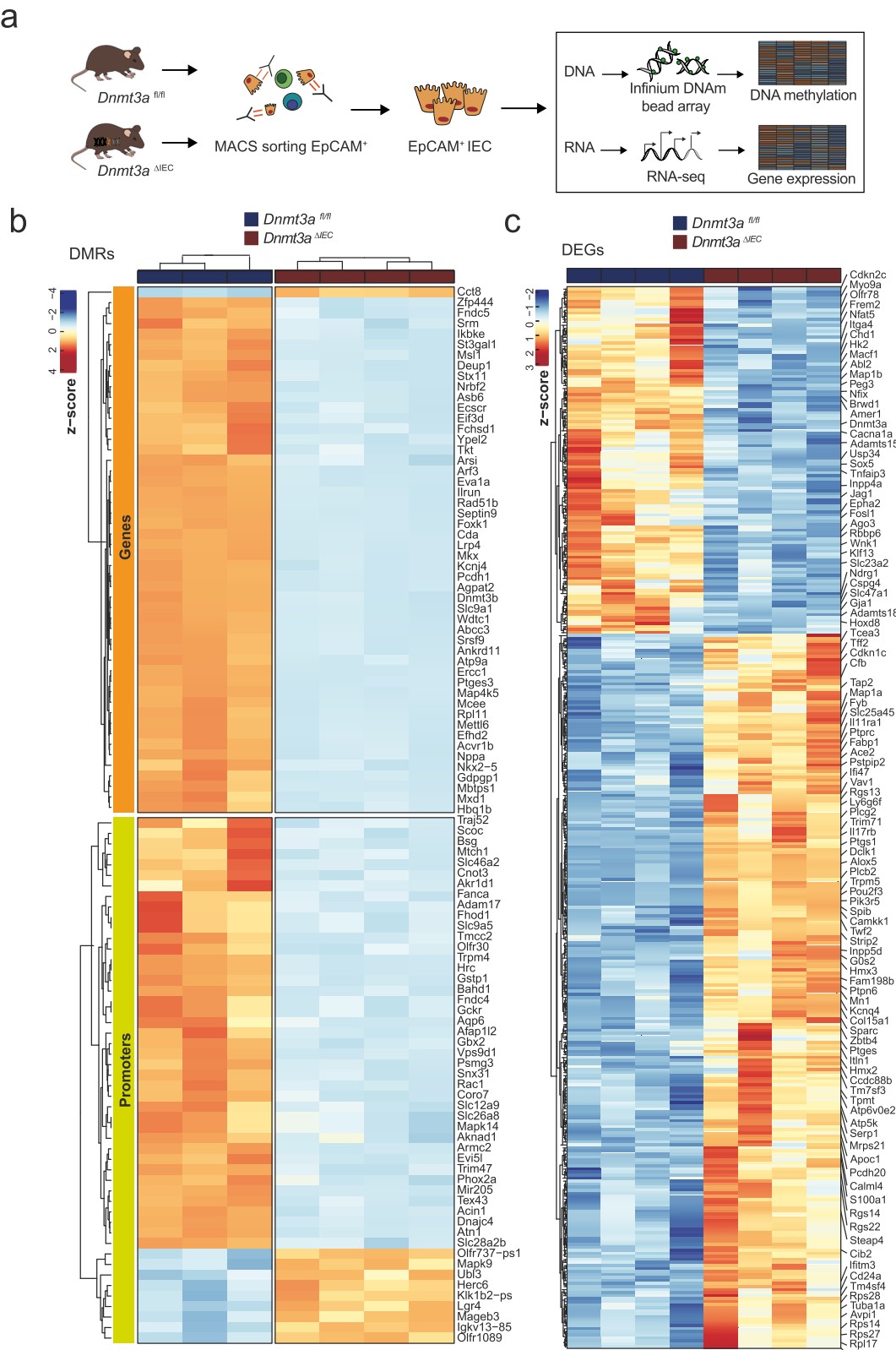

**Fig. 4 | Transcriptome and methylome profiles in *Dnmt3a*-deficient IEC in vivo.** **a** Summary of experimental setup. Graphical elements modified from refs. 17, 74. EpCAM positive IECs were obtained from colon tissues and used for RNA-sequencing and methylation profiling using Infinium Mouse Methylation BeadChip array. Heatmaps showing DMRs (**b**) and DEGs (**c**) identified in *Dnmt3a^ΔIEC^* versus *Dnmt3a^fl/fl^* IECs. (*n* = 4). Scaled gene expression/methylation intensity across all samples are plotted.

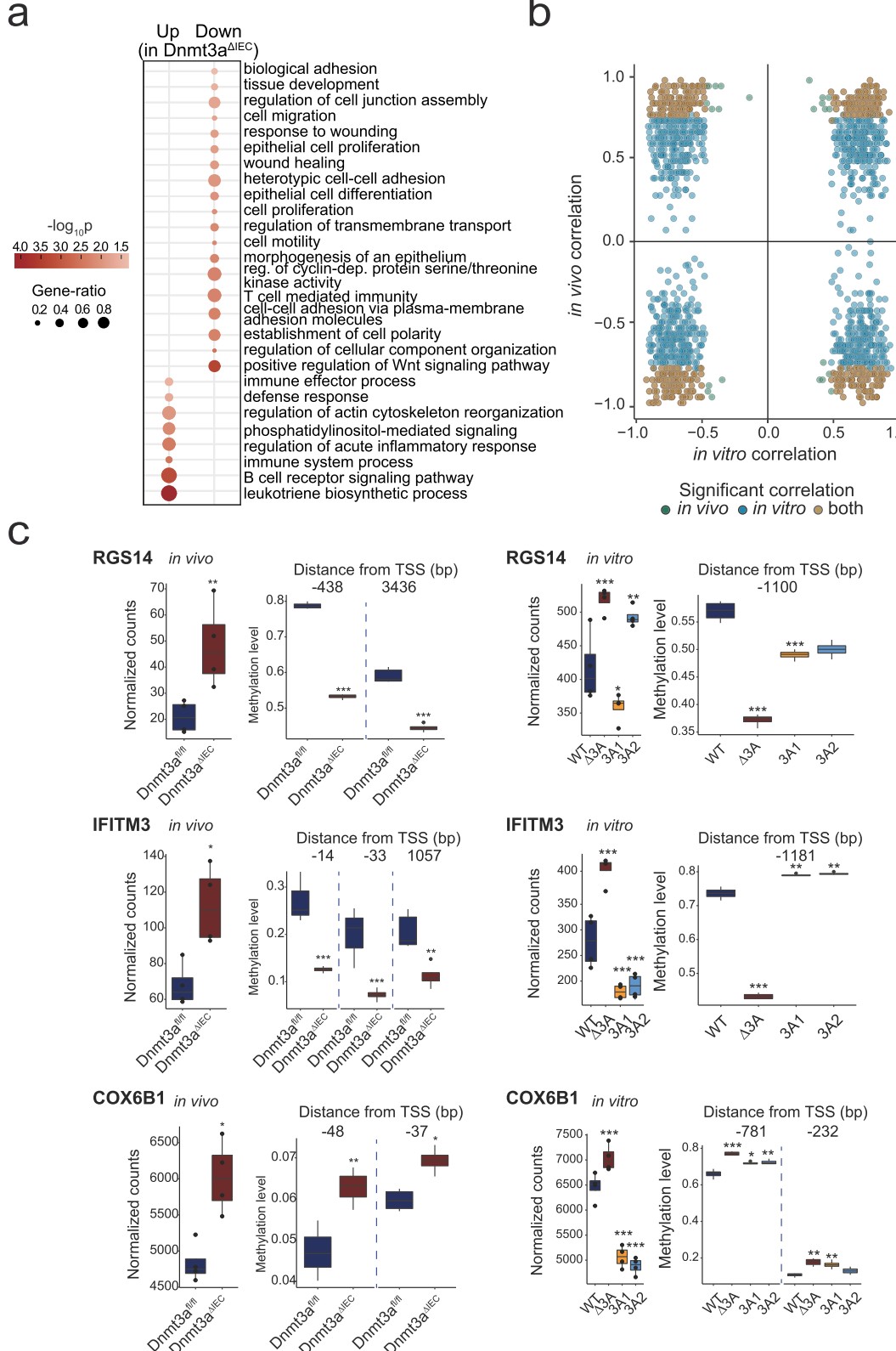

**Fig. 5 | Intersection analysis of DNAm-linked DEGs in vivo and in vitro models.** **a** Gene ontology analysis for biological processes enriched in DEGs in *Dnmt3a*$^{\Delta IEC}$ and *Dnmt3a*$^{fl/fl}$ mice. Dot size is proportional to the gene ratio and color corresponds to the *p*-value of enrichment (two-sided). Top selected GO terms are shown. **b** Comparison of transcription-methylation correlation between in vitro and in vivo models. Blue and green dots represent the rescued genes that are correlated with one or more nearby DMPs in vitro and in vivo, respectively. Brown dots represent the rescued genes that are correlated with one or more nearby DMPs in both models. **c** Common genes regulated by DNMT3A in Caco-2 model (in vitro) and in mouse model (in vivo). Gene expression (left panels) (*n* = 4) and DNA methylation levels (right panels) (*n* = 4) of *RGS14 (Rgs14)*, *IFITM3 (Ifitm3)*, and *COX6B1 (Cox6b1)* are shown. Within each boxplot, the horizontal lines denote the median values and the boxes extend from the 25th to the 75th quartile of the distribution. The vertical lines extend to the most extreme values within 1.5 interquartile range of the 25th and 75th percentile of the distribution. *P*-values are calculated using the DESeq2 method for the transcriptomic data and the hierarchical linear model (limma) for the methylation data. \**p* < 0.05, \*\**p* < 0.01, \*\*\**p* < 0.001, \*\*\*\**p* < 0.0001.

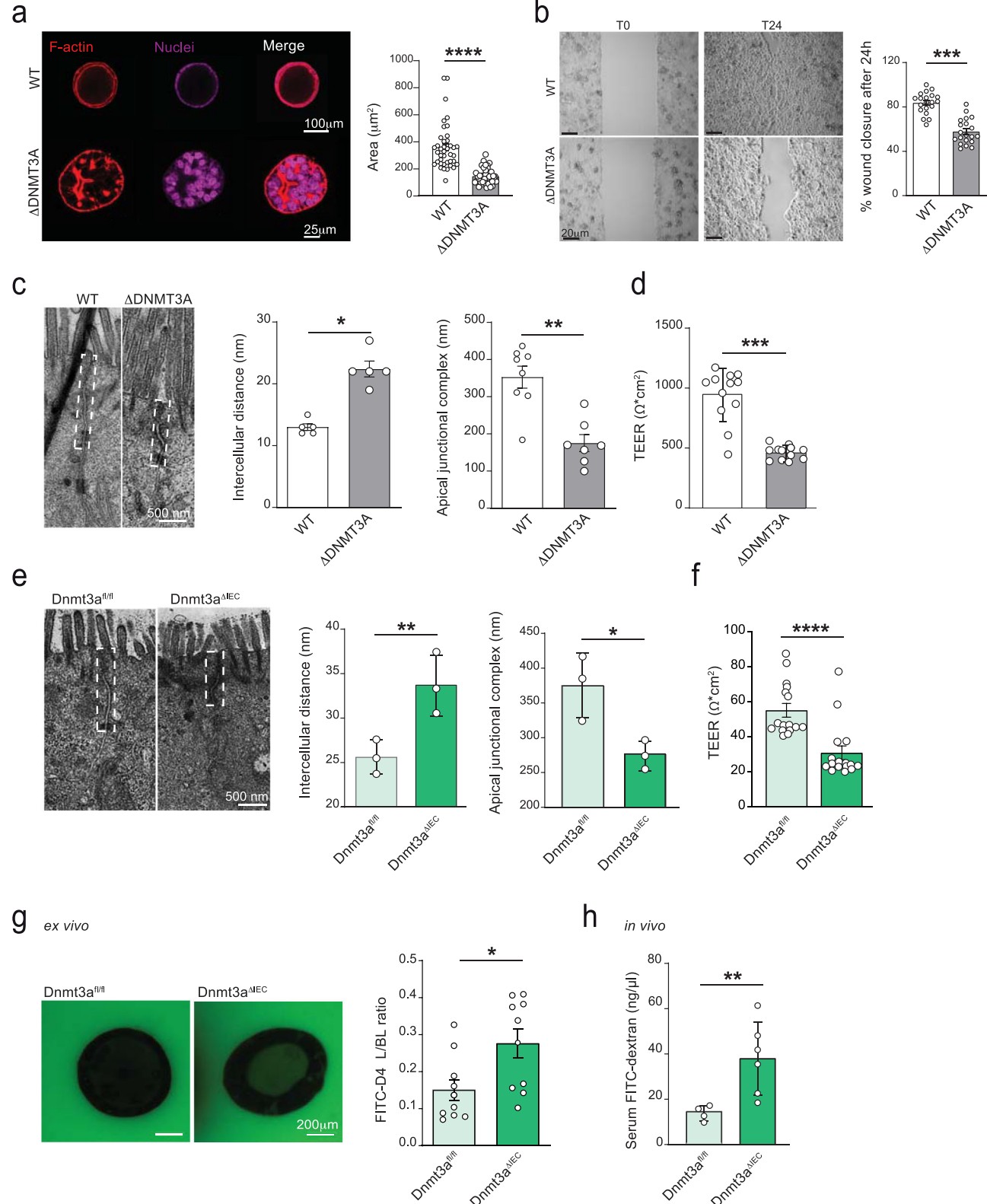

*Dnmt3a*[fl/fl] littermates (Fig. 7c). Based on this experiment, where we observed a delay in body weight gain during the first recovery phase (day 5–10), we hypothesized that DNMT3A plays a critical role during the respective post-colitis recovery phases. Therefore, we next treated the mice only with a short course (5 days) of DSS in drinking water followed by normal drinking water until mice fully recovered their body weight (Fig. 7d). *Dnmt3a*[ΔIEC] mice were characterized by a significant increase in fecal blood at day 5 (Fig. 7e) and reduced intestinal

epithelial cell proliferation rate on day 12 (as measured by BrdU-positive cells) (Fig. 7f) without increased cell death as assessed by TUNEL staining (Supplementary Fig. 5a). To pinpoint altered epigenetic regulatory processes in *Dnmt3a*[ΔIEC] after DSS colitis, we performed genome-wide methylation profiling of crypts isolated from colon tissue of *Dnmt3a*[fl/fl] and *Dnmt3a*[ΔIEC] mice 5 and 12 days after the start of the DSS treatment. Global hypomethylation as a feature of *Dnmt3a*[ΔIEC] compared to *Dnmt3a*[fl/fl] epithelial cells was still observed

**Fig. 6 | DNMT3A regulates intestinal paracellular permeability. a** Representative images of ΔDNMT3A and WT spheroids cultured in 3D matrix for 14 days. Confocal microscopy was used to analyze spheroid organization. Phalloidin (red) and DRAQ5 (purple) was used to visualize actin filaments and nuclei, respectively. Spheroids area (μm²) was determined in WT and ΔDNMT3A ($n = 40$ over 3 independent experiments). **b** Representative images of Caco-2 cells, WT and ΔDNMT3A, at 0 h and 24 h after inducing the gap. The percentage of wound closure after 24 h was measured with a phase-contrast microscope using the absolute wound area normalized to the corresponding value at 0 h. ($n = 20$ over 3 independent experiments). **c** Transmission electron microscopy (TEM) was used to measure intercellular distance and apical-junctional complex length in WT and ΔDNMT3A Caco-2 cells cultivated on a transwell support ($n = 8$). **d** Trans-epithelial electrical resistance (TEER) measurements of differentiated ΔDNMT3A and WT Caco-2 cells ($n = 12$ over 3 independent experiments). **e** Representative TEM images and quantification of intercellular distance and apical-junctional complex length of small intestinal sections from $Dnmt3a^{\Delta IEC}$ and $Dnmt3a^{fl/fl}$ mice ($n = 3$). **f** TEER measurements of colonic organoids from $Dnmt3a^{\Delta IEC}$ and $Dnmt3a^{fl/fl}$ mice grown as monolayer on transwell support ($n = 16$ over 4 independent experiments). **g** Representative images and fluorescence intensity ratio between the luminal (L) and basolateral (BL) side of $Dnmt3a^{\Delta IEC}$ and $Dnmt3a^{fl/fl}$ colonic organoids incubated with FITC-D4 for 24 h ($n = 10$). **h** FITC-dextran quantification in serum isolated from $Dnmt3a^{\Delta IEC}$ and $Dnmt3a^{fl/fl}$ mice ($n = 6$) 1 h after oral administration. The values represent mean ± SEM. Statistical analysis was performed using two-tailed $t$-test with Mann–Whitney correction. *$p < 0.05$, **$p < 0.01$, ***$p < 0.001$, ****$p < 0.0001$.

after DSS treatment (Supplementary Fig. 5b). On comparing the methylation profile of epithelial crypts isolated at day 5 and day 12 of DSS treatment to the baseline in $Dnmt3a^{fl/fl}$ and $Dnmt3a^{\Delta IEC}$ mice, we observed shared and unique changes in methylation in both models. Genes related to promoters differentially methylated uniquely in $Dnmt3a^{\Delta IEC}$ mice (e.g., *Cdkn2c*, *Ccn5*, *Ctnnbip1*, and *Sfrp5*) at day 5 after the initiation of DSS colitis were enriched in processes related to cell junction and cell proliferation (Fig. 7g, Supplementary Data 13). At day 12, processes uniquely enriched in $Dnmt3a^{\Delta IEC}$ mice comprised translation and nitrogen compound biosynthesis. It is important to note that many of the observed methylation changes in $Dnmt3a^{\Delta IEC}$ epithelial crypts represent increased DNAm (Fig. 7g), including hypermethylation of the Muc2 promoter at both timepoints, pointing to complex regulatory changes by mechanisms other than DNMT3A activity after DSS treatment.

To further investigate the reduced regeneration of the intestinal lining in $Dnmt3a^{\Delta IEC}$ mice, we assessed several candidate markers. We observed reduced expression of the intestinal stem cell marker *Lgr5* and the mucosal wound healing marker *Tff3* in $Dnmt3a^{\Delta IEC}$ mice compared to $Dnmt3a^{fl/fl}$ littermates (Fig. 7h). The canonical Wnt signaling marker *Axin2* was downregulated in $Dnmt3a^{\Delta IEC}$ mice during the regenerative phase (Fig. 7h), indicating repression of Wnt activity, which may partially explain the reduced healing response observed in $Dnmt3a^{\Delta IEC}$ mice[21].

We thus addressed epithelial cell type changes and barrier function after DSS challenge in more detail. $Dnmt3a^{\Delta IEC}$ mice during inflammation presented a further decreased number of Goblet cells (identified by PAS staining) (Supplementary Fig. 5c) and downregulation of the Goblet cell marker *Muc2* gene (Fig. 7h).

We also observed a significant downregulation of *Atoh1* and *Notch1* in $Dnmt3a^{\Delta IEC}$ mice during the recovery phase (Supplementary Fig. 5d), as markers for secretory progenitor and crypt proliferative cells, respectively[25]. In addition, we measured mRNA expression levels of the antimicrobial epithelial genes *Lcn2* and *Duox2* in mice from day 5 (acute phase DSS) and day 12 (recovery phase). Lcn2 and Duox2 are downregulated in $Dnmt3a^{\Delta IEC}$ mice compared to the WT animals during the recovery phase (Supplementary Fig. 5e).

We next investigated whether, during chronic intestinal inflammation, key players of the junctional complex are transcriptionally dysregulated. We found that during chronic inflammation, the levels of transcripts encoding the adherens junctional proteins E-cadherin (Cdh1) and β-catenin (Ctnnb1), and the tight junctional proteins occludin (Ocln) and Zonula occludens-1 (ZO-1) are downregulated in $Dnmt3a^{\Delta IEC}$ mice compared to their littermate control (Fig. 8a). We could confirm the downregulation of ZO-1 protein in colonic tissue of $Dnmt3a^{\Delta IEC}$ mice by immunofluorescent staining (Fig. 8b). Here, a reduced and less regular staining pattern of ZO-1 at the apical-junctional complex was found in $Dnmt3a^{\Delta IEC}$ mice challenged with chronic exposure of DSS, which clearly differed from $Dnmt3a^{fl/fl}$ mice. The pattern was also absent in $Dnmt3a^{\Delta IEC}$ mice at baseline.

In acute colitis, we found reduced expression of *Ocln*, *Cdh1*, and *Ctnnb1* mRNA levels compared to their littermate control at day 12 (Fig. 8c), again indicating incomplete restitution of the epithelial layer. We finally used a cell-based TLR4 reporter assay to test for LPS activity in the serum of DSS-treated mice as a proxy for intestinal barrier function in vivo. We stimulated HEK-blue TLR4- secreted alkaline phosphatase (SEAP) reporter cells with a serum derived from untreated $Dnmt3a^{\Delta IEC}$ and $Dnmt3a^{fl/fl}$ mice, as well as from mice at day 5 and 12 after acute DSS challenge (Fig. 8d). Serum derived from $Dnmt3a^{\Delta IEC}$ mice at day 12 led to significantly higher SEAP activity compared with $Dnmt3a^{fl/fl}$ littermates (Fig. 8d). In addition to the permeability defect at baseline, the results suggest a decreased restoration of barrier function in $Dnmt3a$-deficient mice in the recovery phase after an epithelial insult, which may be linked to enhanced inflammatory signals.

## Discussion

DNA methylation (DNAm) is involved in a broad spectrum of biological processes such as cellular development, differentiation, and tissue homeostasis[26,27]. Impaired regulation of DNAm has been associated with numerous pathological conditions, including chronic inflammatory diseases[5,7,28]. However, our understanding of the role of defective DNAm in chronic inflammation remains limited. The genetic association of the de novo methyltransferase enzyme DNMT3A with IBD, and with CD, in particular, suggests a causative role of this part of the epigenetic machinery[11]. Notably, we observed a general downregulation of DNMT3A transcript as well as protein levels in biopsies and purified IECs from patients with IBD and also demonstrate that the genetic risk for IBD at the human DNMT3A locus is mediated by repression of DNMT3A mRNA levels.

Loss-of-function studies of DNMT3A so far had been limited to embryonic stem cells, where DNMT3A deletion resulted in viable pluripotent cell lines, which displayed progressively altered DNA methylation patterns and showed a role of DNMT3-mediated methylation for endoderm differentiation[29,30]. In our comparative study using a human CRISPR-modified IEC cell line (Caco-2) and conditional deletion in mice, we observe specific changes of DNAm signatures and gene expression driven by DNMT3A deletion in differentiated adult intestinal epithelial cells. As expected, we found global hypomethylation as a shared feature of deficient DNMT3A function. Interestingly, we also observed a small proportion of genes/CpG islands that are hypermethylated upon deletion of DNMT3A in both in vitro and in vivo models. Whereas we do not find evidence for compensatory upregulation of the second de novo methyltransferase DNMT3B, the hypermethylated regions might still be a result of altered DNMT3B function as has been described in the previous studies[31–33].

The regulatory effect of DNAm has been canonically linked to interference with the binding of transcription factors in gene promoter regions[34], which in turn leads to transcriptional repression[35]. On the other hand, DNA methylation has also been shown to enhance gene transcription. Indeed, gene body methylation has been positively correlated with expression[36]. In our study, we found that

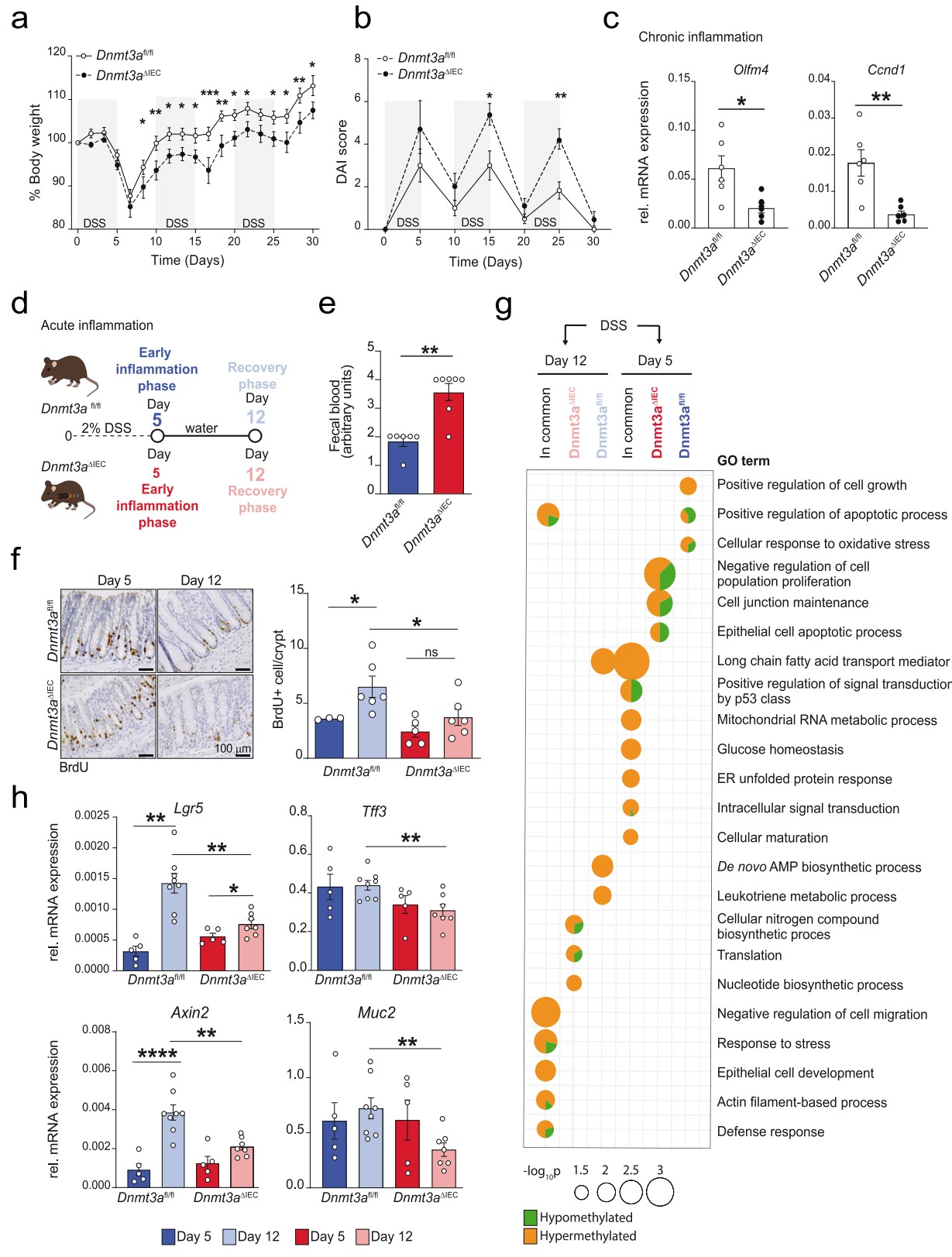

approximately half of the methylation changes, which were positionally linked to a differentially expressed transcript (e.g. COX6B1), follow a such non-canonical pattern, which recapitulates findings from a previous study in human IBD biopsies[7].

In the human in vitro model, hypomethylated positions were enriched in upregulated DEGs related to positive regulation of type I interferon signaling, response to TNF, and intestinal absorption.

Downregulated DNAm-linked DEGs related to the regulation of Wnt signaling, extracellular matrix organization, and epithelial cell morphogenesis were associated with both hypo- and hypermethylated positions. In the purified IECs from Dnmt3a^ΔIEC mice in vivo, we confirmed the downregulation of transcripts related to Wnt signaling, wound healing, and cell migration/proliferation, while upregulated DNAm-linked transcripts were again related to immune and

**Fig. 7 | Conditional deletion of Dnmt3a in IECs increases susceptibility to inflammation.** Chronic colitis was induced by cyclic administration of 2% DSS (*n* = 6 female animals/group). Weight loss (**a**) and diseases activity index (DAI) score (**b**) were monitored every other day until day 30. **c** Colon mRNA expression levels of proliferative markers *Olfm4* and *Ccnd1* in crypts isolated from *Dnmt3a*<sup>ΔIEC</sup> and *Dnmt3a*<sup>fl/fl</sup> mice (qRT-PCR) in chronic DSS colitis at day 30 of the experiment. Murine beta-actin was used as housekeeping gene (*n* = 6). **d** Acute inflammation experimental model workflow (early inflammation group (day 5) *n* = 5, recovery post-DSS group (day 12) *n* = 8). Graphical elements modified from ref. 74. **e** Fecal blood content on day 5 (*n* = 6/7). **f** Representative images and quantification of BrdU-positive cells from colonic sections of *Dnmt3a*<sup>ΔIEC</sup> and *Dnmt3a*<sup>fl/fl</sup> mice. A minimum of 100 crypts/section were assessed. Each dot represents each animal

(*n* = 3/5 day 5, *n* = 6 day 12). **g** Gene ontology (GO) analysis for processes enriched in differentially methylated promoters 5 and 12 days after DSS treatment in Dnmt3a<sup>ΔIEC</sup> and Dnmt3a<sup>fl/fl</sup> mice. Dot size is proportional to the statistical significance of the enrichment with larger dots being more significant and color corresponds to the proportion of hypo- (green) and hypermethylated (orange) promoters contributing to each GO term. Top selected GO terms that are unique to or shared between Dnmt3a<sup>ΔIEC</sup> and Dnmt3a<sup>fl/fl</sup> mice are shown. **h** Colon *Lgr5, Tff3, Axin2, Muc2* mRNA expression levels (qRT-PCR) in crypts isolated from Dnmt3a<sup>ΔIEC</sup> and Dnmt3a<sup>fl/fl</sup> mice on day 5 (*n* = 5) and day 12 (*n* = 7/8). Murine beta-actin was used as housekeeping gene. The values represent mean ± SEM. Statistical analysis was performed using two-tailed *t*-test with Mann−Whitney correction or one-way ANOVA together with Tukey post hoc test. *$p < 0.05$, **$p < 0.01$, ***$p < 0.001$, ****$p < 0.0001$.

inflammatory responses. Strikingly, similar results were obtained in a study investigating purified IECs from pediatric, therapy naïve IBD patients[37].

IBD patients present with intestinal barrier dysfunction[38,39], attributed to dysregulation of proteins involved in epithelial junctional organization[40,41], e.g., characterized by low expression of desmosomal β-catenin and E-cadherin[42,43] as well as to a more penetrable mucus layer[44]. Tight junction function and ultrastructure are altered in patients with IBD[38], which is associated with altered paracellular permeability. Increased epithelial permeability is predictive of relapse in patients with quiescent IBD[45,46] and has long been known to occur in first-degree relatives of patients with Crohn's disease[47]. Indeed, these results were recently confirmed in prospective cohorts, where increased permeability markers preceded the onset of IBD, in particular of Crohn´s disease, by several months[48]. The mechanism behind this lasting functional shift of the intestinal barrier remains poorly understood. In both of our DNMT3A-deficient IEC models (Caco-2 cells and conditional mice), we see an increased intercellular space and shortened cellular apical-junctional complex, phenocopying similar observations in IBD patients[38]. We observed that mice lacking DNMT3A are characterized by increased intestinal permeability, reduced number of Goblet cells, reduced mucus thickness, and increased bacterial DNA in mesenteric lymph nodes (MLNs). These effects are present at baseline and are aggravated during DSS colitis, in particular in the late recovery phase. Here, we showed LPS elevation in the serum of *Dnmt3a*<sup>ΔIEC</sup> mice, as a proxy of delayed and/or inadequate barrier repair after an inflammatory insult.

Intestinal barrier function requires an adequate regenerative response of the epithelium, particularly upon damage. Impaired epithelial regeneration is commonly observed in IBD[49,50], and epithelial growth-promoting principles such as topic EGF enemas or IL-22Fc have been employed in clinical trials in IBD[51,52]. Our study shows that intact DNMT3A is necessary for appropriate epithelial regeneration in vitro and in vivo. Using a wound healing assay in CRISPR-deleted Caco-2 monolayers in vitro and DSS challenge in vivo as models for epithelial damage, we show that loss of DNMT3A is associated with a delayed regeneration process, indicated by a decreased cell proliferation rate and impaired migration. A lack of induction of protective antimicrobial epithelial transcripts (e.g., *Lcn2* and *Duox2*) may further contribute to delayed mucosal repair in *Dnmt3a*<sup>ΔIEC</sup> mice.

Moreover, both in the in vitro and the in vivo model, GO analysis revealed deregulation of the Wnt pathway. Functionally, deletion of DNMT3A also leads to diminished Wnt signaling, indicated by decreased activity in a TCF/Wnt-dependent reporter assay. A previous study has found that adequate Wnt signals are needed for intestinal Goblet cell differentiation[53], suggesting that diminished Goblet cell numbers in *Dnmt3a*<sup>ΔIEC</sup> mice might also be linked to epigenetically impaired Wnt signals. In the recovery phase of the DSS model, we found downregulation of *Axin1* levels and reduced *Atoh1* expression as a marker for the secretory lineage. Our data thus provide evidence for epigenetic control of this pathway, which—if perturbed—is associated with increased susceptibility to

inflammation in a murine model of inflammation; however, it must be noted that the influence of aberrant Wnt signals in IBD is complex and far from being understood[54–56].

Limitations of our study include that—while we observe a specific downregulation of DNMT3A in IECs from IBD patients- for reasons of informed consent, we cannot directly link this effect to the genetic background of the respective patient population. For inference of the genetic effect on DNMT3A mRNA expression, we used a well-characterized, large blood-based mRNAseq dataset instead[14], but we cannot rule out that additional tissue-specific effects may modulate the genetic control of DNMT3A expression. While we clearly demonstrate increased intestinal permeability at baseline in vitro and in vivo, for the DSS colitis we relied on LPS serum measurements by TLR4 reporter cells as an indicator for a leaky barrier. Although partial equivalence of the two methods has been shown, we thus cannot formally compare the baseline permeability with DSS-induced changes. Undetectable levels of LPS at baseline in both WT and *Dnmt3a*<sup>ΔIEC</sup> mice and sustained elevation in the recovery phase after DSS only in *Dnmt3a*<sup>ΔIEC</sup> mice, however, clearly support the notion of an impaired barrier restoration in *Dnmt3a*<sup>ΔIEC</sup> mice upon an inflammatory insult. Thus, although we find increased bacterial DNA in the mesenteric lymph nodes of *Dnmt3a*<sup>ΔIEC</sup> mice and demonstrate elevated serum LPS activity after DSS colitis, we want to point out that this finding does not prove relevant translocation of viable bacteria. Finally, as purification is notoriously difficult under inflammatory conditions, we cannot completely rule out that the complex shifts of DNAm patterns in epithelial fractions of mice after DSS treatment, which also include hypermethylated sites, are partially linked to infiltrating immune cells.

Collectively, our data reveal a functional role of epithelial DNMT3A in maintaining homeostasis and controlling inflammatory response in the intestinal mucosa. Future studies will have to elucidate how a general defect of the methylation machinery can lead to a deregulation of a distinct set of genes in a specific adult tissue compartment such as the intestinal epithelium. In line with earlier findings demonstrating the impact of microbial signals on DNMT3A expression and DNAm signatures in IECs[57], our results may help define potential early therapeutic intervention points in individuals at risk for IBD, e.g., by rationale ecobiotherapy aiming to normalize DNAm signatures.

## Methods

All research complied with relevant ethical regulations, and the human study protocol was approved by the ethics committee of the Medical Faculty at Kiel University. Animal studies conformed to ethical rules of handling laboratory animals, and all procedures were approved prior to the study by the committee for animal welfare of the state of Schleswig-Holstein.

### Patient recruitment

Adult patients with confirmed IBD were recruited between 2013 and 2016, where active inflammation was assessed endoscopically. The independent biopsy cohort for transcriptional analysis

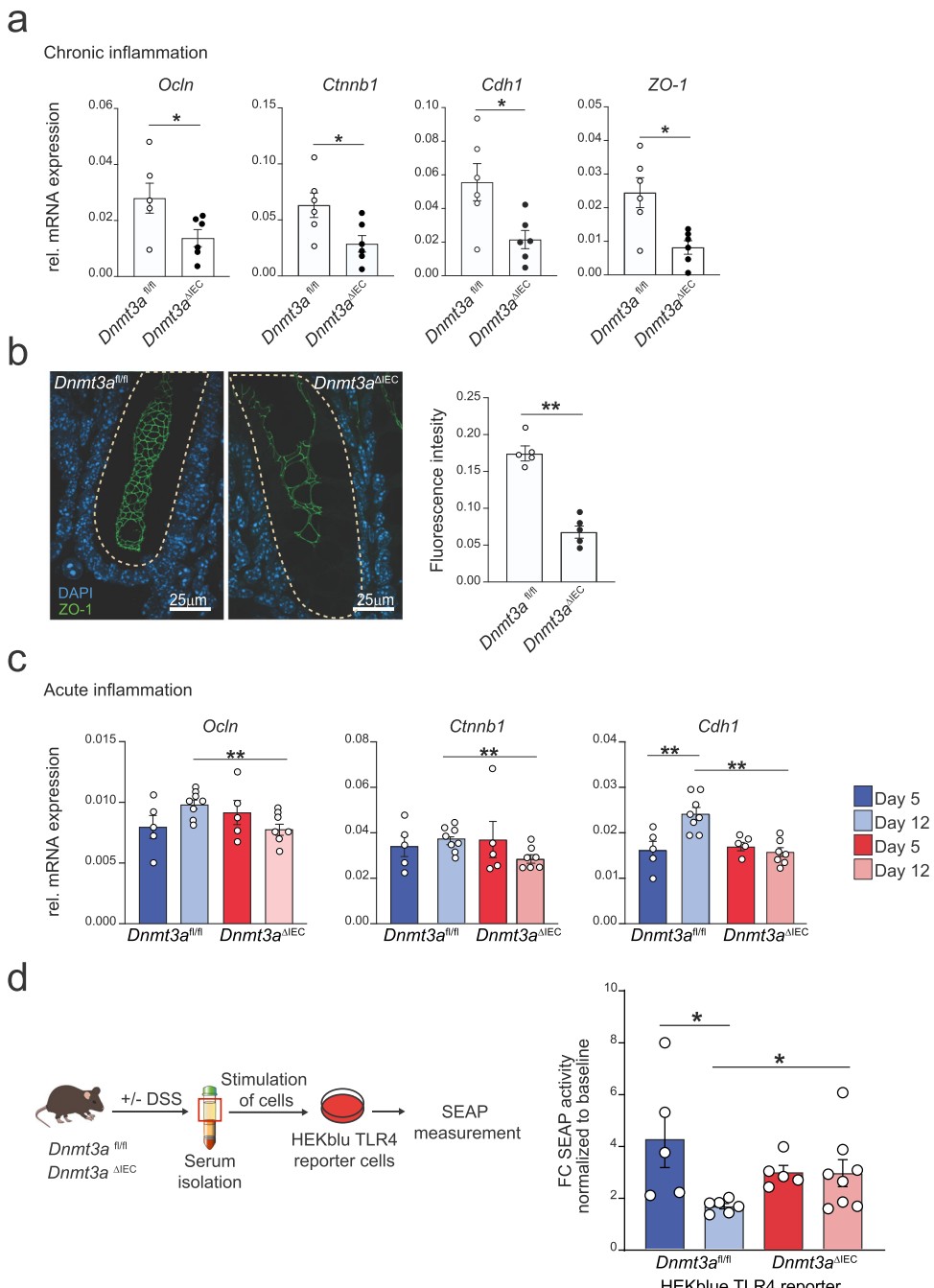

**Fig. 8 | Apical-junctional proteins are downregulated in *Dnmt3a*$^{\Delta IEC}$ mice during inflammation. a** Colon mRNA expression levels of *Ocln, Ctnnb1, Cdh1,* and *ZO-1* in crypts isolated from *Dnmt3a*$^{\Delta IEC}$ and *Dnmt3a*$^{fl/fl}$ mice (qRT-PCR) in chronic DSS colitis at day 30 of the experiment (*n* = 6). Murine beta-actin was used as housekeeping gene. **b** Representative images and quantification of ZO-1 fluorescence intensity from colonic tissue sections of *Dnmt3a*$^{\Delta IEC}$ and *Dnmt3a*$^{fl/fl}$ mice in chronic DSS colitis at day 30 of the experiment. ZO-1 is depicted in green while nuclei in light blue. Each dot represents each animal (*n* = 5). **c** *Ocln, Ctnnb1,* and *Cdh1* mRNA expression levels in crypts isolated from colon tissues derived from *Dnmt3a*$^{\Delta IEC}$ and *Dnmt3a*$^{fl/fl}$ mice during acute inflammation (day 5, *n* = 5) and

recovery phase (day 12, *n* = 7/8). Murine beta-actin was used as housekeeping gene. **d** Schematic workflow (left panel) and SEAP production (right panel) of HEK-Blue TLR4 reporter assay. HEK-Blue TLR4 reporter cells were stimulated with serum derived from mice subjected to early inflammation (day 5, *n* = 5) and recovery phase (day 12, *n* = 6/8). Data are shown as fold change normalized with serum from untreated animals. Each dot represents a different mouse donor. The values represent mean ± SEM. Graphical elements modified from refs. 17, 74. Statistical analysis was performed using two-tailed *t*-test with Mann–Whitney correction (**a**, **b**) or one-way ANOVA together with Tukey post hoc test (**c**, **d**). **p* < 0.05, ***p* < 0.01, ****p* < 0.001, *****p* < 0.0001.

comprised 150 unrelated individuals (*n* = 30 active UC, *n* = 30 inactive UC; *n* = 30 active CD, *n* = 30 inactive CD; *n* = 30 healthy individuals). Healthy controls were recruited from colonic cancer surveillance colonoscopies or other indications (e.g., unclear weight loss) and were examined by the same endoscopists. Individuals were considered healthy when no pathological findings

were made as part of the endoscopic examination. All biopsies have been taken at the same site in the sigmoidal colon (-25 cm from the anal verge), the definition active reflects that biopsies were taken from an affected area, but not from the ground of ulceration. The IBD patient cohort was a heterogenous group with different treatments, reflecting the clinical scenario of a tertiary

referral center. Approval was granted by the ethics committee of the medical faculty of Kiel University before the study (B231/98), and written informed consent forms were obtained from each patient.

## Cell culture and transfection
Human colonic carcinoma Caco-2 (ACC 169) cells were purchased from the German Collection of Microorganisms and Cell Cultures (DSMZ) and maintained in MEM medium containing 20% (vol/vol) FCS at 37 °C with 5% $CO_2$. Cells were cultured in respective media supplemented with 20% (vol/vol) FCS without antibiotics. For 3D epithelial cell culture, Caco-2 cells were grown in Matrigel®(BD, Biosciences)[19]. For functional barrier analysis, Caco-2 cells were seeded on transwell inserts (0.4 µm pore size, TH Geyer) and maintained for 14–21 days. THP-1 cells (ACC16) were purchased from DSMZ and cultured in RPMI medium containing 10% FCS at 37 °C with 5% $CO_2$. Transfections were performed after 24 h post-seeding using Viromer BLUE transfection reagent (Lipocalyx) for siRNA and Lipofectamin®3000 (ThermoFisher Scientific) for plasmid DNA according to the manufacturer´s instructions.

## DNA construct
A CRISPR/Cas9 vector was created according to the manufacturer's instructions, by ligation of a dsDNA oligo targeting DNMT3A1 and DNMT3A2 (5′-GAGGACCGAAAGGTGAGCAGGTTTT-3′) into pCRISPR-CD4 (Thermo Fisher Scientific). For rescue experiments, DNMT3A1 and DNMT3A2 isoforms were fused to a TY1 tag and cloned into PB-CMV-MCS-EF1α-GreenPuro (BioCat GmbH).

## Cell line generation
A DNMT3A-deficient Caco-2 cell line was created by transient transfection of pCRISPR-CD4 using Lipofectamine 3000 according to the manufacturer's instructions (Thermo Fisher Scientific). After 24 h, cells expressing CD4 were isolated with the Geneart CD4 Enrichment Kit (Thermo Fisher Scientific) and serially diluted in 96-well plates to obtain clonal cell lines. After clonal expansion, ablation of DNMT3A was assessed by western blotting. For rescue experiments, DNMT3A-knockout cells were co-transfected with TY1-DNMT3A-Short/long isoforms (in PB-CMV-MCS-EF1α-Green-Puro) and Super piggyBac Transposase (BioCat GmbH) using Lipofectamine 3000 (Thermo Fisher Scientific). Cells were serially diluted in 96-well plate and clonally expanded in the presence of puromycin. Re-expression of DNMT3A short/long isoforms was validated by Western blotting.

## Isolation of crypts and cultivation of murine colonic organoids
Crypts were isolated from the mouse colon by EDTA-based $Ca^{2+}$/ $Mg^{2+}$ chelation, and intestinal organoids were cultivated as described by Sato et al.[58]. In brief, the colon was removed and cut longitudinally. Intestinal pieces were incubated in cold PBS and washed by gently shaking. The supernatant was removed, and intestinal pieces were incubated in cold fresh 2.5 mM EDTA-PBS solution for 40 min at 4 °C. Supernatant was removed, and fresh cold PBS was added. Crypts were detached by mechanical brake using a serological pipette. The crypt suspension was collected. This procedure was repeated three times. The crypt suspension was collected. This procedure was repeated three times. The crypt suspension was passed through a 100 µm strainer and centrifuged at $300 \times g$ at 4 °C. Epithelial crypts were resuspended in Matrigel (BD Bioscience) and embedded in 24-well plates and cultivated in L-WRN conditioned medium. The medium was changed twice per week, and organoids were used after 5 days of cultivation. For stimulation experiments, organoids were seeded in a 24-well plate and stimulated with TNFa (50 ng/ml), IFNg (50 ng/ml), LPS (50 ng/ml) in L-WRN medium for 24 h.

## Establishment of human intestinal organoids
Human intestinal biopsy samples were obtained from patients (Crohn's disease and ulcerative colitis) who underwent endoscopic examination. The study was approved by the Ethics Committee of the Medical Faculty, University Kiel (vote B231/98/13), and written informed consent was obtained from each patient prior to study-related procedures. Isolation of the crypts and the subsequent establishment of intestinal organoids were performed as previously described[59]. Briefly, biopsy samples were incubated in 2.5 and 15 mM EDTA for the colon and small intestine, respectively, and crypts were collected. Isolated crypts were embedded in Matrigel and placed in 24-well culture plate. Crypts were maintained in 50% L-WRN conditioned medium, generated as previously described using the L-WRN cell line[60], supplemented with recombinant human EGF (50 ng/ml, PeproTech), Y-27632 (10 µM, Sigma–Aldrich), A83-01 (500 nM, Tocris), Nicotinamide (10 mM, Sigma–Aldrich), N2 supplement and B12 supplement (Thermo Scientific) and SB202190 (10 µM, Enzo).

## Permeability assay in vitro
For paracellular permeability, Caco-2 cells were plated on a transwell insert for 14 days. Lucifer Yellow (LY, Sigma) was added to the apical compartment. From the basolateral compartment, 50 µl sample was transferred in a 96-well plate, and the fluorescence was quantified using a fluorescence microtiter plate reader at 428/549 nm. For 3D permeability assay, colonic organoid or Caco-2 spheroid-like structures were incubated with 4 kDa fluorescein isothiocyanate-labeled dextran (FITC-4D, Sigma) for 24 h. The flux of FITC-4D from the basal to the luminal compartment (L/B ratio) was assessed using confocal microscopy[19]. Confocal images were taken with Leica laser scanning confocal microscope (Leica Microsystems GmbH) and processed using ImageJ software[61].

## Promoter-mediated luciferase reporTER ASSAy
For Wnt/b-catenin transcriptional activation, M50 Super 8× TOPFlash reporter assay was used. The reporter contains height copies of TCF/LEF binding sites upstream of luciferase. M50 Super 8× TOPFlash was a gift from Randall Moon (Addgene plasmid #12456; http://n2t.net/addgene:12456; RRID:Addgene_12456)[62]. Bioluminescence from the Renilla luciferase was used as an internal control for transfection efficiency and cell viability.

## In vitro wound healing assay
Caco-2 (WT and ΔDNMT3A) cells were plated on four-well culture insert (©ibidi GmbH, Martinsried, Germany) at cell density of $6 \times 10^{-5}$ cells/ml. After 24 h the insert was removed, and the cells were washed twice with PBS. Images of the gap were taken at time 0, immediately after removing the insert, and at time 24 h using an inverted microscope (Zeiss Axio Vert.A1) and processed with AxioVision Imaging software (Carl Zeiss MicroImaging Inc.).

## Co-culture system with THP-1 cells
For co-culture experiments, Caco-2 were seeded on a transwell for 2 weeks, as described above. THP-1 cells were differentiated with PMA (100 nM) for 24 h and pre-stimulated with LPS (10 ng/ul) and INF gamma (10 ng/ul) for 4 h to induce an inflamed co-culture model. Afterward, transwells containing Caco-2 cells were transferred to THP-1 differentiated activated cells. The medium was replaced every other day, and barrier integrity was monitored every day by TEER measurements.

## Immunoblotting
For protein extraction of organoids, Matrigel was dissolved using Cell recovery solution (BD, Bioscience) followed by lysis using SDS-based DLB buffer + 1% protease and phosphatase inhibitors. Samples were

heated at 95 °C for 5 min followed by sonification for 5 s two times. Lysates were centrifugated at 16,000 × *g* for 15 min at 4 °C, and the supernatant was collected. Proteins were separated on 10% poly-acrylamide gels by SDS-PAGE and transferred onto polyvinylidene fluoride membranes (Millipore). Antibodies targeting DNMT3A (1:1000) (R&D Systems) and GAPDH (1:1000) and ACTB (1:1000) (Santa Cruz) were used.

## Transmission electron microscopy (TEM)

Monolayer differentiated Caco-2 cells, and intestinal tissue samples were fixed in 3% glutaraldehyde buffer (3% Glutaraldehyde in 0.1 M phosphate buffer, pH 7.4), samples were post-fixed in 2% Osmium tetroxide and then embedded into araldite using standard proce-dures. After ultrathin sectioning, samples were stained using uranyl-acetate and then inserted into a JEOL1400 Plus TEM (JEOL, Munich, Germany) operating at 100 kV, and images were recorded on a F416 digital camera (TVIPS, Munich, Germany)[63]. To ensure sagittal sec-tions through the cells, microvilli were used as reference points. Only positions where the entire microvillus was detected were imaged for quantification. Measurements of the adherens junctions or cellular contact zones were performed in EM MENU (TVIPS, Munich Germany).

## RNA extracts and gene expression analysis

After total RNA extraction using the RNeasy kit (Qiagen), 1 μg of RNA was reverse-transcribed to cDNA using Maxima H Minus First Strand cDNA Synthesis kit (Thermo Scientific) according to the manu-facturer's instructions. Quantitative Real-Time PCR was performed using the TaqMan Gene Expression Master Mix (Applied Biosystems) according to the manufacturer´s protocol and analyzed by the 7900HT Fast Real Time PCR System (Applied Biosystems). Murine TaqMan assays used: *Dnmt3a* (01027166), *Dnmt3b* (01240113), *Olfm4* (01320260), *Ccnd1* (00432359), *Lgr5* (00438905), *Tff3* (00495590), *Axin2* (00443610), *Muc2* (00458299), *Ocln* (00500912), *Ctnnb1* (00483039), *Cdh1* (01247357), *Notch1* (00627185), *Atoh1* (00476035), *Lcn2* (01324470), *Duox2* (01326247), *Cxcl10* (00445235), *Cxcl1* (00433859), *Tnfa* (00443258). Human TaqMan assays used: *DNMT3A* (01027166), *DNMT3B* (00171876), *DNMT1* (00945875). Human primer used (SYBER green qRT-PCR): *CXCL10* (F: CCTTTCCCATCTTCCAAGGGT, R: GGAGGATGGCAGTGGAAGTC), *CCND1* (F: ACTACCGCCTCACACGCT TC, R: CAGGTCCACCTCCTCCTCCT).

Relative transcript levels were determined using Actb and Gapdh as housekeepers.

## Transcriptomics analysis

RNA samples from Caco-2 cells were sequenced on HiSeq3000 (Illumina, San Diego, United States) (2 ×75 bp) while that from IECs from *Dnmt3a*^fl/fl^ and *Dnmt3a*^ΔIEC^ were sequenced NovaSeq 6000 (2 ×50 bp) using Illumina total RNA stranded TruSeq protocol. An in-house RNA-seq pipeline was used to map and align the sequenced data (https://github.com/nf-core/rnaseq). Adapters and low-quality bases from the RNA-seq reads were removed using Trim Galore (version 0.4.4) (http://www.bioinformatics.babraham.ac.uk/projects/trim_galore/). Reads that were shorter than 35 bp after trimming were discarded. The filtered reads were mapped to the human genome (GRCh38, gencode version 25) or mouse genome (GRCm38) using STAR aligner (version 2.5.2b)[64]. featureCounts (version 1.5.2) was used to estimate the expression counts of the genes[65]. The expression counts were normalized across samples using the DESeq normalization method. Differentially expressed genes were identified using the Bioconductor package DESeq2 (ver-sion 1.20.0)[66]. Genes with FDR-adjusted *p*-value of less than 0.05 were regarded as differentially expressed (DEGs).

## DNA methylation analysis

DNA was isolated from Caco-2 cells and purified IECs using DNeasy Blood and Tissue Kit (Qiagen) according to the manufacturer´s instructions. DNA methylation data from Infinium© MethylationEPIC BeadChip and Infinium© Mouse Methylation BeadChip were analyzed using the Bioconductor package RnBeads (version 1.12.1)[67]. Sites that overlapped with SNPs and had unreliable measurements were filtered out. Context-specific probes, probes on the sex chromosomes, and probes with missing values were also removed. In total, 41,693 out of 866,895 probes for Caco-2 cells and 54,392 out of 261,600 probes for mouse intestinal tissue were filtered. The signal intensity values were normalized using the dasen method. Differentially methylated posi-tions (DMPs) and regions (DMRs) between WT and Δ3 A, 3A1 and 3A2 cells, *Dnmt3a*^fl/fl^, and *Dnmt3a*^ΔIEC^ mice and before and after DSS treat-ment were identified using the automatically selected rank cutoff of RnBeads.

## DNA methylation-transcriptome integrated analysis

For the integrated analysis of gene expression with DNA methyla-tion, we first identified DMPs located 5000 bp upstream and down-stream of the transcription start sites of DEGs. Spearman's rank correlation coefficient between the normalized expression count of each DEG and the methylation intensity of its corresponding DMPs was calculated. To test the statistical significance of the correlations, we calculated the false discovery rate (FDR) using a permutation approach.

## Functional enrichment analysis

All gene ontology enrichment analyses were conducted using the Bioconductor package topGO (version 2.32.0)[68], with genes with similar expression levels as the universe set. In the topGO analysis, the Fisher.elim *p*-value, calculated using the weight algorithm, of 0.05 was used as the significance threshold.

## Transcriptome-wide association analysis (TWAS) using IBD GWAS summary statistics

The TWAS approach can be viewed as an expression imputation and transcriptome-wide screening method to test for gene-disease asso-ciations with GWAS datasets. We performed gene expression imputa-tion (machine learning models for genes via elastic net) and transcriptome-wide association analysis for 11,475 genes with S-PrediXcan[69] using two GWAS summary statistics from the Interna-tional IBD Genetics Consortium (https://ibdgenetics.org/; 5956 CD cases and 14,927 controls as well as 6968 UC cases and 20,464 controls[15]) in combination with whole-blood RNA-seq and genome-wide genotype reference data for 922 individuals from the DGN cohort[70,71]. Estimated expression levels were tested for association with disease using logistic regression in S-PrediXcan. From TWAS associa-tion results, we extracted all imputable genes (m = 44) from the known IBD risk locus 2p23.3[14,15] (extended region chr2:24,000−27,900 kb to capture long-range *cis*-eQTL effects) to prioritize candidate genes at 2p23.3.

## DNA isolation and bacterial quantification

DNA was extracted using Dneasy Power Soil Pro (Qiagen) kit according to the manufacturer's instructions. Real-time PCR ampli-fication was performed using 16 S universal primers (F: ACT CCT ACG GGA GGC AG, R: GAC TAC CAG GGT ATC TAA TCC) and probe (CAG CAG CCG CGG TA) that target the V3−V4 region of the bacterial 16 S ribosomal gene. qPCR performed in duplicate on a VIIA 7 PCR system (ThermoFisher, Waltham, MA). The absolute number of 16 S gene copies was determined by comparison with a standard curve gener-ated by serial dilution of E.coli 16 S rDNA. A total 16 S rRNA gene count was normalized by mg of sample.

## Mice

*Dnmt3a*$^{fl/fl}$ mice (C57BL /6 J background) were generated by a commercial supplier (EMMA). In brief, exons 13–17 were flanked by LoxP sites to enable its excision by Cre recombinase. Upstream exon 13, a distal loxP site was introduced with an FRT flanked neomycin selection cassette. The resultant mouse line was bred with Flp deleter-mice (C57BL/6-Tg(CAG-Flpe)2Arte, Taconic) constitutively expressing Flp recombinase to remove the neomycin selection cassette, creating a *Dnmt3a*$^{fl/fl}$ mouse in which *Dnmt3a* exons 13–17 were flanked by two loxP sites. These mice were crossed with Vil1-Cre deleter-mice (C57BL/6JxSJL/J background), resulting in *Villin(V)-cre*$^{+/−}$;*Dnmt3a*$^{fl/fl}$ mice with intestinal-epithelial-cell-specific *Dnmt3a* deletion (*Dnmt3a*$^{ΔIEC}$). For the baseline investigations male and female mice (age 8–20 weeks) were employed at an ~1:1 ratio, distributed equally between KO and WT groups. For the DSS experiments, only female mice were used. All mice were maintained in a specific pathogen-free facility. Mice were provided with food and water *ad libitum* and maintained in a 12 h light-dark cycle at 23 °C of ambient temperature with 60 ± 5% humidity. Mice were euthanized by cervical dislocation prior to sampling organs for histological and molecular analyses. Procedures involving animals were conducted in conformity with national and international laws and policies, and all procedures were approved prior to the study by the committee for animal welfare of the state of Schleswig-Holstein (acceptance no.: V242-7224.121-33).

## In vivo treatment of mice

For chronic colitis induction, *Dnmt3a*$^{fl/fl}$ and *Dnmt3a*$^{ΔIEC}$ mice (6 females/group, age 8–10 weeks) were supplied with 2% DSS (MP Biomedicals) dissolved in autoclaved drinking water for 5 days followed by 5 days of regular drinking water; this cycle was repeated three times, resulting in a 30-day experimental period. For early-stage colitis induction, 13 females/group mice (age 8–10 weeks) were subjected to 2% DSS for 5 days followed by 6 days of regular drinking water. The Disease Activity Index was obtained[72]. These animal experiments were approved by the committee for animal welfare of the state of Schleswig-Holstein (acceptance no.: V242-17696/2016 (32-3/16)).

## Immunohistochemistry and immunofluorescence

For immunohistochemical staining, 5 μm sections of paraffin-embedded colon swiss rolls were deparaffinized with Xylol substitute (Roth), incubated in citrate buffer for 3 min, and subsequently blocked with blocking serum (Vectorlab, Peterborough,UK) for 20 min. Primary anti-BrdU (1:10 dilution, BD Pharmingen, Heidelberg, Germany) and anti-DNMT3A (1:100 dilution, Cell Signaling, Cambridge, UK) were incubated overnight. Sections were washed and incubated with secondary antibody goat anti-mouse immunoglobulin G (1:1000; Jackson ImmunoResearch, Ely, UK) and DAB substrate with Vectastain ABC Kit (Vectorlab, Peterborough,UK). For TUNEL assay, slides were subjected to Apop Tag Plus Peroxidase In situ Apoptosis Detection kit (Merck Millipore) according to the manufacturer's protocol. For PAS staining, slides were deparaffinized and hydrated. Oxidation was performed in 0.5% periodic acid solution for 5 min. Slides were stained with Schiff reagent for 15 min and counterstained for 1 min. Slides were dehydrated and mounted using xylene-based mounting media. For ZO-1 staining, slides were incubated with Anti-Zonula occludens-1 (1:100, Invitrogen 40-2200) overnight at 4 °C. Slides were incubated with secondary antibody and DAPI (1:20,000) (Sigma–Aldrich, D9542) for nuclei staining. For FISH assay, colon tissues were hybridized and stained with the probe (EUB338_Cy3, Microsynth, 3913816). After washing and blocking in 5% BSA, samples were incubated with primary antibody (Anti-E-cadherin 1:100 dilution, BD Biosciences 610182) overnight. Sections were washed and incubated with secondary antibodies (Alexa Fluor™ Plus 488-conjugated) and DAPI (1:20,000) (Sigma–Aldrich, D9542) for

nuclei staining. Images were taken with a fluorescence microscope (Zeiss AxioImager.Z1) and processed with ZEN 3.4 Blue edition software.

For immunofluorescence staining, cells were washed, fixed in 4% PFA-PBS, and permeabilized in 0.5% Triton X-100-PBS. The unspecific binding size was blocked in 5% BSA-PBS and incubated with primary antibody overnight at 4 °C and secondary antibody for 1 h. DRAQ5 (1:5000) and Phalloidin (1:200) were used for DNA counterstaining and F-Actin, respectively. Confocal images were taken with Leica laser scanning confocal microscope (Leica Microsystems GmbH) and processed using ImageJ software.

## FITC-dextran permeability assay in vivo

*Dnmt3a*$^{fl/fl}$ and *Dnmt3a*$^{ΔIEC}$ mice fasted for 4 h and 4-kDa FITC dextran (Sigma–Aldrich, 60 mg/100 g body weight) was applied orally. Mice were euthanized after 1 h, and blood was collected into gel-containing tubes. The serum was isolated by centrifugation at $10,000 × g$ for 5 min and diluted 1:1 with PBS. Fluorescence was measured on a spectrophotometer in 96-well plates at 528 nm. FITC-dextran concentrations were calculated with the use of a standard curve (concentrations prepared in PBS ranging from 0 to 800 μg/mL 4-kDa FITC dextran). Background signals in the serum of mice receiving PBS were subtracted from those mice treated with the 4 kDa. The experiments were approved prior to the study by the committee for animal welfare of the state of Schleswig-Holstein (Approval no. V242-28372/2022 (29-4/22)).

## TLR4 Bioassay (Intestinal permeability in vivo)

HEK-Blue-TLR4 reporter cell line (cat.hkb-htlr4,Invivogen) was used to measure the presence of LPS released in the serum. This reporter cell line stably expressed human TLR4 receptor and a secreted alkaline phosphatase (SEAP) reporter gene under the control of an IL12 p40 minimal promoter fused to NF$\kappa$B and AP-1 binding sites. Cells were maintained according to the manufacturer's protocol. 10 ml of serum from untreated and DSS-treated mice were added to a flat bottom of a 96-well plate. HEK cell suspension (140,000 cells/ml) resuspended in HEK-Blue detection medium (Invivogen) was added to each well. Cells were incubated with samples overnight at 37 °C in 5% $CO_2$, and plates were read on a spectrophotometer at 620 nm.

## Flow cytometry

Immune cells isolated from colonic lamina propria or mLN were washed, and Fc-Block (CD16/CD32) was performed on ice for 15 min. Antibody incubation was performed at 4 °C in the dark. Antibody incubation was performed using the following dilutions CD45-FITC (BioLegend) 1:100, NK1.1-PE (eBioscience) 1:80, CD3e-AF532 (eBioscience), CD11b-PerCP/Cy5.5 (Biolegend) 1:50, CD4-BV711 (BioLegend) 1:200, CD19-BV605 (BioLegend) 1:100, CD8-BV785 (BioLegend) 1:100, CD11c-Pacific Blue (BioLegend) 1:50. Cell viability was assesses using Zombie Red™ Fixable Viability Kit Biolegend 1:1000. Data were acquired using SA3800 Spectral Analyzer (Sony Biotechnology) and were analyzed using SA3800 Software version 2.0.5.54250.

## Statistics

Statistical analysis was performed using the GraphPad Prism 9 software package (GraphPad Software Inc., La Jolla, USA). For comparisons, Student's *t*-test and one-way ANOVA with Mann–Whitney/ Tukey/Bonferroni post-test were performed. Data are shown as mean ± standard error of the mean (SEM). A *p*-value of ≤0.05 was considered as significant (*). A *p*-value of ≤0.01 was considered as strongly significant (**) and *p*-value of ≤0.001 as highly significant (***).

## Reporting summary

Further information on research design is available in the Nature Research Reporting Summary linked to this article.

## Data availability

All data generated or analyzed during this study are available from the corresponding author on reasonable request. The raw and processed RNA-sequencing data generated in this study have been deposited in the NCBI GEO database under the accession number GSE210714. The Infinium© MethylationEPIC BeadChip and Infinium© Mouse Methylation BeadChip array data generated in this study have been deposited in the NCBI GEO database under accession code GSE210721. Supplementary Figure 1d was generated using published data[14]. Source data are provided with this paper.

## Code availability

The custom codes used in this study are shared on Github (https://github.com/Systems-Immunology-IKMB/DNMT3A_epithelial)[73].

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

## Acknowledgements

We gratefully appreciate the technical assistance of Sabine Kock, Maren Reffelmann, Stefanie Baumgarten, Dorina Ölsner, Karina Greve, Melanie Nebendahl, Tatjana Schmidtke, and Vivian Wegener. This work was supported by the Research Training Group "Genes, Environment and Inflammation", supported by the Deutsche Forschungsgemeinschaft (RTG 1743/2), by the ExC 2167 PMI RTFIII and TI1, the DFG RG miTarget (P3 and P5), CRC1182 C2, project SO1141/10-1, EU SYSCID H2020 #733100 and the EU IMI2 network 3TR, WP7 (831434). The funding bodies had no part or influence on the design of the study and data collection, analysis, or interpretation.

## Author contributions

A.Fa. and P.R. designed the study; A.Fa., D.B., J.K., S.W.S., S.T.St., P.A., D.E., G.I., F.T., B.M., A.H., J.P.B., R.H., A.L., S.I., F.H., A.Fr., S.H., S.N., K.A., S.Sc., F.S., and G.N. performed experiments and analyzed data; N.M. analyzed data; A.Fa., N.M., and P.R. wrote the manuscript.

## Funding

## Competing interests

The authors declare no competing interests.

## Additional information

[1]Institute of Clinical Molecular Biology, Christian-Albrechts-University and University Hospital Schleswig-Holstein, Campus Kiel, 24105 Kiel, Germany. [2]Institute of Functional and Clinical Anatomy, Friedrich-Alexander-University Erlangen-Nürnberg (FAU), 91058 Erlangen, Germany. [3]Advanced Research Institute, Tokyo Medical and Dental University, Tokyo, Japan. [4]Department of Internal Medicine I., Christian-Albrechts-University and University Hospital Schleswig-Holstein, Campus Kiel, 24105 Kiel, Germany. [5]Department of Dermatology and Allergy, University Hospital Schleswig-Holstein, Campus Kiel, Kiel, Germany. [6]I. Department of Medicine, University Medical Center Hamburg-Eppendorf, Hamburg, Germany. [7]Department of Experimental Oncology, European Institute of Oncology IRCCS (IEO), 20139 Milan, Italy. [8]These authors contributed equally: Antonella Fazio, Dora Bordoni. [9]These authors jointly supervised this work: Neha Mishra, Philip Rosenstiel. ✉e-mail: p.rosenstiel@mucosa.de

