## [Peer Review File · Nature Communications]

DNA methyltransferase 3A controls intestinal epithelial barrier function and regeneration in the colonREVIEWER COMMENTS

Reviewer #1 (Remarks to the Author):

Review of Fazio et al. "DNA methyltransferase 3A controls intestinal epithelial barrier function and regeneration in the colon".

This manuscript describes DNMT3A as a key player in maintaining intestinal homeostasis and gut barrier function. The authors performed genetic ablation of DNMT3A in a human colon cancer cell line Caco2 and in intestinal epithelial cells (IECs) in mice. They showed that loss of functional DNMT3A led to differential changes of gene expression and DNA methylation. The authors then performed intestinal permeability analysis and DSS-induced inflammation experiments. They found altered intestinal ultrastructure changes and increased susceptibility to DSS-induced colitis. The authors concluded (abstract) "a critical role for DNMT3A in orchestrating intestinal epithelial homeostasis and response to tissue damage and suggest an involvement of impaired epithelial DNMT3A function in the etiology of IBD."

Epigenetic dysregulation is recognized to be critical for pathogenesis of inflammatory bowel disease (IBD). Therefore, the topic of this study is of great interest to a broad research community. In its current state, however, the manuscript describing these findings is poorly detailed, and as such, it is not entirely convincing. Specific comments follow.

1. The authors stated in the abstract that "genetic variants in the DNMT3A locus have been associated with IBD", but never described them in detail. For example, where is the reference? Do the patients analyzed in this study contain DNMT3A mutations or variants?
2. There is a concern with the statistical analysis. For example, it was stated in the Methods that two-way ANOVA was performed, but in Figure 1, it appears that the statistical analysis was done using one-way ANOVA.
3. The results in Fig. 1 are difficult to follow and interpret. The authors never described the patient samples in detail. For example, were the CD and UC samples collected before or after treatment? Did the organoids grow and proliferate at similar rate among three groups (healthy vs. non inflamed vs. inflamed)? There was no significant difference found between healthy vs. non-inflamed CD organoids. Any explanation? Finally, what is the justification for exposing murine organoids to pro-inflammatory stimuli? Why not the human organoids? And why not comparing the healthy vs CD organoids?
4. In Fig. 2, it is unclear why 850K EPIC array is chosen since the authors described RRBS in mouse IECs. It is also unclear how the data were combined and compared. Have the authors confirmed the analysis? It is unexpected that after DNMT3A deletion, methylation at CpG islands increases (Fig. 2c). Any explanation? Based on the correlation plots, I would expect strong correlation between expression and methylation changes, but the examples shown in Suppl. Fig. 1 do not clearly support this. This is a major concern because the authors should perform detailed epigenetic analysis to validate the correlation between DNA methylation and expression. This would allow them to draw the conclusion in a more meaningful way.
5. Similarly, the results in Fig. 3 are difficult to follow and interpret. For example, the gene names are shown for expression changes but not for methylation changes (b and c). Fig. 3f does not clearly support the statement "same direction of correlation between DEG and DMP". The expression of Rgs14 in IECs was minimal compared to Caco2, and the methylation changes were inconsistent between human and mouse. Similarly, no significant change of Adam15 expression was found in Dnmt3a deleted IECs.
6. Fig. 4: The changes of intercellular distance and apical-junctional measurements are relatively subtle. The authors should perform standard assays to validate their findings, i.e., the Ussing chamber with paracellular flux and electrical measurements and in vivo FITC-dextran assays for permeability.
7. Fig. 5: the DSS experiments were focused on the analysis of stem cell and proliferative markers. As such, it lacks mechanistic insights, particularly regarding epigenetic regulation. The authors mentioned about TFF3 expression changes, have they looked the methylation changes?
8. It was suggested that only during chronic inflammation the tight junctional proteins were significantly downregulated in Dnmt3a deleted mice. But after a close look at the data (Fig. 6c), the significance was mainly driven by one outlier. Have the authors repeated the experiments?

Reviewer #2 (Remarks to the Author):

Fazio and collaborators provide relevant and novel data on the role of DNA methyltransferase 3A (DNMT3A) on intestinal epithelial cell function.

DNMTs mediate DNA methylation and are responsible for the expression or repression of hundreds of genes. A polymorphism in DNMT3A has been related to the risk of developing inflammatory bowel disease thus suggesting a potential link between the function of this methylation process and chronic intestinal inflammation. Nonetheless, little information is currently available on the involvement of DNMT3A-mediated methylation in disease manifestations. Thus, the study by Fazio and col. is relevant to the field as it provides novel information on how DNMT3A participates in epithelial cell homeostasis and adequate response to injury.

Our understanding of how IBD becomes a chronic inflammatory condition is still rather incomplete.

One potential mechanism, as the authors explain is epigenetic modifications that can alter gene expression long term and cause disease to become chronic. In addition, whether dependent or independent of epigenetic changes, induction of memory immune responses could also represent an important pathway driving disease chronic manifestations. Understanding how these or other mechanisms can result in life-long sustained disease manifestation is a key unanswered question, relevant to finding treatments that can change the disease's natural history.

Overall, the manuscript is well written, the figures are clear and well explained. In addition, the methodology used is adequate and includes patient and murine derive tissue and cells, and in vivo animal models. Using CRISP deletion of DNMT3A they elegantly demonstrate that depletion of this gene leads to the regulation of thousands of genes in epithelial cells.

I only have a few comments and suggestions that I hope can further improve the quality of the study.

The authors focus on the role of DNMT3A on epithelial cells. While I agree that it is the right approach for their study, it should be mentioned in the manuscript that this methyltransferase is expressed by several other cell types. Indeed, DNMT3A has been shown to play important role in immune cells (including macrophages, T and B cells) that are relevant to mucosal homeostasis. Thus, when looking at the expression of DNMT3A in whole biopsies from IBD patients, the potential contribution of other cells to the expression of the gene (Fig 1a) should be considered and mentioned.

I am curious to know if the authors have also observed the regulation of DNMT3A by TNF using human organoids. Human organoids in previous studies showed very little responses to TNF, while they are very sensitive to other cytokines (i.e. IFN γ or IL-22 for example). Also, what other genes were regulated together with DNMT3A in their experiments with TNF and murine organoids?

It would be important to describe if DNMT3A AIEC mice given their increased epithelial permeability, show any baseline changes in mucosal immune cell composition and/or phenotype. Information on IEL, mucosal macrophages, and Ig responses would provide important information to the reader. Even if no changes are seen, that is also information that should be added to further understand the implications of DNMT3A function on mucosal homeostasis.

Minor comments

- In Figure 5c add time point d30?.
- Please provide expression of antimicrobial inflammatory epithelial genes (LCN2, DUOX2, etc).
- Figure 6a, SEAP activity of serum from non-colitic mice (both wt and KO) should be included or used to show data as FC from baseline.

- Data in Fig 6b and c would go better in figure 5.
- Yui et al (ref 44), is not a clinical trial and should not be cited as such.
- Also, ref 42 is a trial on teduglutide a GLP-2, not on IL-22

Reviewer #3 (Remarks to the Author):

The study addresses a very interesting and important question in the field of IBD centering on the role of epigenetic regulation of intestinal epithelial cell function. The authors focus on a DNA methylating enzyme (DNMT3A) and use CRISPR/Cas9 targeting of CACO2 cells to generate an in vitro model comparing it to an in vivo model using the Villin-Cre to drive conditional ablation in the murine intestinal epithelium. In analyzing these models, the authors determine the methylation as well as transcriptional network to determine the processes that DNMT3A controls. The manuscript is very well written with very interesting results. A criticism of the data is interpretation of DSS susceptibility. Intestinal permeability after DSS treatment increases even in mice without primary defects in tight junctions. Delays in inflammatory clearance could also explain a delay in goblet cell reaccumulation. Overall, there are a number of addressable questions that will improve the quality of the study.

Comments and Suggestions on figures:

In figure 1, author shows downregulation of DNMT3A in IBD as well as in response to TNF α . Is this specific to DNMT3A or is it seen with other DNA methylases.

Does Figure 1A reflect one or both isoforms of DNMT3A?

Figure 3:

Authors describe intersection of the CACO2 and in vivo epithelial cell experiments. It would be helpful to see the gene list and degree of expression change perhaps as supplement.

Figure 5

In figure 5C, are these *Olfm4* and *Ccnd1* tested under DSS treatment?

Proliferation is measured. What about cell death?

Are there differences in permeability to insoluble molecules (e.g. FITC-dextran) of the intestine under steady state conditions in vivo? Is this exacerbated under DSS treatment conditions?

The authors show a decrease in some stem cell markers in response to DSS. Is the loss of Goblet cells secondary to a differentiation defect across the secretory lineage? Is there a loss of secretory progenitors?

Figure 6

Authors use HEK-BLUE TLR4 assay to measure LPS after DSS treatment. These differences do not appear very large in magnitude. Is there an increase in bacterial translocation to distant sites (e.g. MLN, spleen, liver) after DSS treatment?

Authors measure mRNA levels of key junctional molecules. Are these perturbed at the protein level?

Authors state that the decrease in PAS positive cells and decreased *Muc2* expression would lead to changes in mucous thickness. Is this true in steady state? No direct measure of mucus layer is presented. Does this result in increased bacterial translocation or tissue bound bacteria?

Reviewer #1 (Remarks to the Author):

Review of Fazio et al. "DNA methyltransferase 3A controls intestinal epithelial barrier function and regeneration in the colon". This manuscript describes DNMT3A as a key player in maintaining intestinal homeostasis and gut barrier function. The authors performed genetic ablation of DNMT3A in a human colon cancer cell line Caco2 and in intestinal epithelial cells (IECs) in mice. They showed that loss of functional DNMT3A led to differential changes of gene expression and DNA methylation. The authors then performed intestinal permeability analysis and DSS-induced inflammation experiments. They found altered intestinal ultrastructure changes and increased susceptibility to DSS-induced colitis. The authors concluded (abstract) "a critical role for DNMT3A in orchestrating intestinal epithelial homeostasis and response to tissue damage and suggest an involvement of impaired epithelial DNMT3A function in the aetiology of IBD."

Epigenetic dysregulation is recognized to be critical for pathogenesis of inflammatory bowel disease (IBD). Therefore, the topic of this study is of great interest to a broad research community. In its current state, however, the manuscript describing these findings is poorly detailed, and as such, it is not entirely convincing.

We thank the reviewer for the detailed and constructive criticism, which helped to improve the manuscript

Specific comments follow.

1. The authors stated in the abstract that "genetic variants in the DNMT3A locus have been associated with IBD", but never described them in detail. For example, where is the reference? Do the patients analysed in this study contain DNMT3A mutations or variants?

We agree that previous knowledge on the genetic association of the DNMT3A locus with IBD, and with CD in particular, had not been described in sufficient detail. We have now inserted appropriate references. Due to consent reasons, we are not able to interrogate the effects of genetic variants in the patient derived mRNA and organoid-based data sets in the study. We have thus taken a three-tiered approach:

- i. We performed gene expression imputation and transcriptome-wide association analysis using the S-PrediXcan algorithm (Gamazin et al., Nat Genet 2015; Barbeira et al., Nat Comm 2018) using GWAS summary statistics of 5,956 CD cases and 14,927 controls as well as 6,968 UC cases and 20,464 controls from the International IBD Genetics Consortium and whole-blood RNA-seq and genome-wide genotype reference data for 922 individuals using reference transcriptome data from the DGN cohort (Battle et al., Genome Research 2014). Imputed reduced DNMT3A expression was significantly associated with genetic risk for CD ($P_{DNMT3A}=2.93 \times 10^{-6}$) and was the only gene at the 2p23.3 risk locus to achieve transcriptome-wide significance (Supplementary Table 3; $P_{Bonferroni}=0.05/11475=4.36 \times 10^{-6}$ for 11,475 imputable genes), indicating that the genetic risk for CD at this locus is mediated via reduced gene expression of DNMT3A) Suppl. Fig1d.
- ii. We increased the number of mRNA-based analyses using an independent larger cohort of IBD patients (refer to point 3 for patients' information) to minimize the risk of stochastic effects (e.g. overrepresentation of risk variants), also to show the specificity of DNMT3A regulation vs. other targets e.g. DNMT1. We do observe a clear downregulation of DNMT3A mRNA levels in both CD and UC, which is further aggravated by disease activity, which indicates a potential effect of inflammation beyond the rather CD-specific genetic risk.
- iii. For the formal genetic expression trait analysis, no reference data set for intestinal epithelial cells was available. As we used a well characterized, large blood-based mRNAseq data set instead, we cannot rule out that additional tissue-specific effects may modulate the genetic control of DNMT3A expression. We clearly note that in the limitations of study section in the

discussion. Still, the analysis to our knowledge is the first to suggest a functional effect of the lead SNP in this important risk region at genome-wide significance.

2. There is a concern with the statistical analysis. For example, it was stated in the Methods that two-way ANOVA was performed, but in Figure 1, it appears that the statistical analysis was done using one-way ANOVA.

We performed statistical analysis using one-way ANOVA for the comparison of gene expression between 3 or more categorical groups (e.g. Healthy controls, CD inflamed, CD non-inflamed). The mention of two-way ANOVA in the method section was indeed a mistake, which has now been corrected.

3. The results in Fig. 1 are difficult to follow and interpret. The authors never described the patient samples in detail. For example, were the CD and UC samples collected before or after treatment?

We have re-arranged the structure of Figure 1 and we repeated expression analyses with biopsy-derived RNA samples from a larger IBD cohort that has been described before in detail in Nikolaus et al., *Gastroenterology* 2017. The independent biopsy cohort for transcriptional analysis comprised 150 unrelated individuals (n = 30 active UC, n = 30 inactive UC; n = 30 active CD, n = 30 inactive CD; n = 30 healthy individuals). Healthy controls were recruited from colonic cancer surveillance colonoscopies or other indications (e.g. unclear weight loss) and were examined by the same endoscopists. Individuals were considered healthy, when no pathological findings were made as part of the endoscopic examination. All biopsies have been taken at the same site in the sigmoidal colon (approximately 25 cm from the anal verge), the definition “active” reflects that biopsies were taken from an affected area, but not from the ground of an ulceration. The IBD patient cohort was a heterogenous group with different treatments, reflecting the clinical scenario of a tertiary referral center. Patient info are in Supplementary Table 1.

Did the organoids grow and proliferate at similar rate among three groups (healthy vs. non inflamed vs. inflamed)?

We measured mRNA expression levels of the proliferative marker CyclinD1 (CCND1), and we observe that IBD organoids (CD non inflamed and CD inflamed) had downregulation of CCND1 compared to healthy control, suggesting reduced proliferation rates. In addition, we measured mRNA expression levels of the pro inflammatory marker CXCL10, and we see upregulation in both IBD group compared to the control, conforming the inflammatory features characteristic of IBD patients. (Fig. 1b).

There was no significant difference found between healthy vs. non-inflamed CD organoids. Any explanation?

We recognize that DNMT3A protein expression is not significantly different between non-inflamed CD patients and healthy controls, which could be a result of biological heterogeneity of individual samples. In such small sample sizes, individual samples can contribute heavily to the overall variability. We would however like to point out that there is a trend of decreased expression of DNMT3A in organoids derived from non-inflamed CD samples as well. Moreover, we measured mRNA expression levels of DNMT3A using qRT-PCR on human colonic organoids. We now show that at mRNA level, DNMT3A is in fact significantly downregulated in inflamed and non-inflamed samples.

Finally, what is the justification for exposing murine organoids to pro-inflammatory stimuli? Why not the human organoids? And why not comparing the healthy vs CD organoids?

In order to exclude genetic heterogeneity and patient variability in our assay, we decided to use murine organoids derived from three individual mice having the same genetic background and exposed them to prototypical inflammatory stimuli, such as TNF or IFN. We clarify the rationale for this approach in the text and also point out that murine vs. human organoids may respond differently to pro-inflammatory stimuli.

4. In Fig. 2, it is unclear why 850K EPIC array is chosen since the authors described RRBS in mouse IECs. It is also unclear how the data were combined and compared. Have the authors confirmed the analysis?

We agree with the reviewer that direct comparison of RRBS and chip-based methylation analysis has several obstacles, e.g. differential representation of methylated regions between the two methods. As our analysis has a focus on gene expression changes *in cis* of differentially methylated positions/regions (i.e. a gene-centric approach), we thus decided to remove the RRBS data set and to re-do the entire analysis with the same DNA sample set from isolated murine intestinal epithelial cells using a newly available murine DNAm array (Infinium Mouse Methylation BeadChip, Illumina). This array has an emphasis on gene-centric features, similar to the human EPIC array and allows a more direct comparison of features and regions. The findings are contained in the new figure 3.

It is unexpected that after DNMT3A deletion, methylation at CpG islands increases (Fig. 2c). Any explanation?

The reviewer raises an important point, which was not properly discussed in the previous version of the manuscript. We observe a proportion of hypermethylated sites/regions upon DNMT3A deletion in the Caco-2 cells in gene bodies and CpG islands. We confirm this pattern in the new array-based murine data set in primary intestinal epithelial cells. These observations are in line with evidence that DNA methylation at CpG islands and at gene bodies of highly expressed genes is a canonical activity described for DNMT3B (Yang et al., 2014; Baubec et al., 2015; Duymich et al., 2016). Therefore, although we could not detect a compensatory upregulation of DNMT3B levels upon DNMT3A deletion, the observed hypermethylation could result from an imbalanced DNMT3B activity at the differentially methylated sites. We added a sentence in the discussion section reflecting on this finding.

Based on the correlation plots, I would expect strong correlation between expression and methylation changes, but the examples shown in Suppl. Fig. 1 do not clearly support this. This is a major concern because the authors should perform detailed epigenetic analysis to validate the correlation between DNA methylation and expression. This would allow them to draw the conclusion in a more meaningful way.

We agree that the prior visualization was partially misleading, as we did not include significance levels of differential methylation and depicted larger genetic regions that also contained unregulated DMPs. We have now depicted examples where (1) DNA methylation changes are linked to differential expression *in cis* and (2) which are shared between mouse (new DNA methylation data) and human samples. We clearly describe how the correlation plot is linked to the examples and discuss the non-canonical example COX6B1, in which hypermethylation upstream of the TSS is linked to higher expression levels both in Caco2 cells as well murine primary intestinal epithelial cells.

5. Similarly, the results in Fig. 3 are difficult to follow and interpret. For example, the gene names are shown for expression changes but not for methylation changes (b and c).

We apologize for the sub-optimal presentation of the data. As stated above, we have re-done the entire DNA methylation analysis, which formed the basis of Figure 3. We included gene names in the heatmap visualization of the methylation analysis (panel b) and provided lists for the updated correlated expression/methylation pairs (mouse vs. human) in the Supplementary Table 7.

Fig. 3f does not clearly support the statement “same direction of correlation between DEG and DMP”. The expression of *Rgs14* in IECs was minimal compared to Caco2, and the methylation changes were inconsistent between human and mouse. Similarly, no significant change of *Adam15* expression was found in *Dnmt3a* deleted IECs.

As stated above, we have performed new correlation analyses starting from the array-based murine DNA methylation data. We again apologize that we did not include statistical significance levels in the previous figure. We, however, respectfully disagree that the relative read count difference for *RGS14* between primary mouse epithelial cells and human Caco2 cell line would imply that the example is not relevant. *RGS14* is in the top 60% of the highest expressed genes in the mouse epithelial cells and hence is not minimally expressed. Moreover, since the mouse IEC samples and the human Caco2 cell samples were treated as two independent experiments and hence not normalized together, a direct comparison between the expression level of a gene in the two datasets cannot be made. We now depict three clear examples where expression changes are correlated to cis-linked DNA methylation variation. 2 of them are canonically linked (*RGS14* and *IFITM3*): lower methylation associated with higher expression in the *DNMT3A/Dnmt3a*-deficient situation, but we also decided to show a non-canonical example (*COX6B1*): higher methylation levels upstream of the TSS are linked to higher expression levels in the deficient human and murine cells (Figure 3f).

6. Fig. 4: The changes of intercellular distance and apical-junctional measurements are relatively subtle. The authors should perform standard assays to validate their findings, i.e., the Ussing chamber with paracellular flux and electrical measurements and *in vivo* FITC-dextran assays for permeability.

We thank the reviewer that additional evidence corroborating the electron-microscopic findings on the apical junctional complex would strengthen the message that impaired *DNMT3A* function affects barrier integrity of the intestinal epithelium. We added new analyses and rearranged the corresponding Figure 4 accordingly. (1) We measured transepithelial resistance in organoid monolayers from *Dnmt3a* fl/fl vs. *Dnmt3a* Δ IEC mice validating the electrophysiological findings from the CRISPR-Caco2 cells (Figure 4f). (2) As suggested by the reviewer we performed an *in vivo* FITC dextran permeability assay by oral gavage and subsequent measurement of serum concentration levels. In this set-up, we now clearly demonstrate that there is increased permeability at baseline (Figure 4h).

7. Fig. 5: the DSS experiments were focused on the analysis of stem cell and proliferative markers. As such, it lacks mechanistic insights, particularly regarding epigenetic regulation. The authors mentioned about *TFF3* expression changes, have they looked the methylation changes?

We have performed additional array-based DNA methylation analyses on intestinal tissue from acute DSS colitis in order to highlight common and unique methylation changes between *Dnmt3a* fl/fl vs. *Dnmt3a* Δ IEC mice at day 5 (early) and day 12 (late) inflammation (Figure 5h, Suppl. Fig. 5b and Supplementary Table 15). The analysis shows that differentially methylated promoters belong to the genes that are enriched in processes related to around cell junction maintenance and regulation of proliferation uniquely in the *Dnmt3a* Δ IEC mice. Among the marker genes that are dysregulated after DSS treatment in *Dnmt3a* Δ IEC mice, *Muc2* promoter was also found to be hypermethylated upon DSS treatment and was included in the enriched GO term “negative regulation of cell proliferation”.

8. It was suggested that only during chronic inflammation the tight junctional proteins were significantly downregulated in *Dnmt3a* deleted mice. But after a close look at the data (Fig. 6c), the significance was mainly driven by one outlier. Have the authors repeated the experiments?

We thank the reviewer for the comment. We have performed ROUT outlier test for outlier identification and no such outliers have been identified, thus we kindly disagree that the signal is spurious, although we agree that the variation of gene expression is higher in the Dnmt3a fl/fl (WT) mice. We furthermore see the same regulation in the acute model at day 12 (recovery phase) vs. chronic model. In both models, appropriate numbers of mice have been employed to demonstrate the statistically significant biological phenotype, thus by animal protection laws we would not be allowed to repeat the exact same experiment. In addition, we performed immunofluorescence quantification of ZO1 in colon sections of mice from the chronic DSS-induced colitis. In agreement with the gene expression data Dnmt3a Δ IEC mice showed a reduced fluorescence intensity and architectural distortion of ZO1 immunoreactivity. (Figure 6b)

Reviewer #2 (Remarks to the Author):

Fazio and collaborators provide relevant and novel data on the role of DNA methyltransferase 3A (DNMT3A) on intestinal epithelial cell function. DNMTs mediate DNA methylation and are responsible for the expression or repression of hundreds of genes. A polymorphism in DNMT3A has been related to the risk of developing inflammatory bowel disease thus suggesting a potential link between the function of this methylation process and chronic intestinal inflammation. Nonetheless, little information is currently available on the involvement of DNMT3A-mediated methylation in disease manifestations. Thus, the study by Fazio and col. is relevant to the field as it provides novel information on how DNMT3A participates in epithelial cell homeostasis and adequate response to injury. Our understanding of how IBD becomes a chronic inflammatory condition is still rather incomplete. One potential mechanism, as the authors explain is epigenetic modifications that can alter gene expression long term and cause disease to become chronic. In addition, whether dependent or independent of epigenetic changes, induction of memory immune responses could also represent an important pathway driving disease chronic manifestations. Understanding how these or other mechanisms can result in life-long sustained disease manifestation is a key unanswered question, relevant to finding treatments that can change the disease's natural history

Overall, the manuscript is well written, the figures are clear and well explained. In addition, the methodology used is adequate and includes patient and murine derive tissue and cells, and in vivo animal models. Using CRISPR deletion of DNMT3A they elegantly demonstrate that depletion of this gene leads to the regulation of thousands of genes in epithelial cells.

We thank the reviewer for the appreciation of our study subject and findings.

I only have a few comments and suggestions that I hope can further improve the quality of the study.

The authors focus on the role of DNMT3A on epithelial cells. While I agree that it is the right approach for their study, it should be mentioned in the manuscript that this methyltransferase is expressed by several other cell types. Indeed, DNMT3A has been shown to play important role in immune cells (including macrophages, T and B cells) that are relevant to mucosal homeostasis. Thus, when looking at the expression of DNMT3A in whole biopsies from IBD patients, the potential contribution of other cells to the expression of the gene (Fig 1a) should be considered and mentioned.

The reviewer raises a salient point, limiting the analysis to a single cell entity, i.e. intestinal epithelial cells does of course not exclude a role of the gene in another cell type. We have now clearly described this in the limitations of study section. We also deem that this discussion is particularly important as the new analysis on the genetic effects of CD risk SNPs in the DNMT3A locus could only be performed in a large reference data set from peripheral blood, where we cannot rule out tissue-specific regulatory effects.

I am curious to know if the authors have also observed the regulation of DNMT3A by TNF using human organoids. Human organoids in previous studies showed very little responses to TNF, while they are very sensitive to other cytokines (i.e., IFN γ or IL-22 for example). Also, what other genes were regulated together with DNMT3A in their experiments with TNF and murine organoids?

We thank the reviewer for raising this point. We did not perform stimulation experiment in human organoids for several reasons. We wanted to rule out any influence of genetic background/patient variability effect on the outcome of the experiment. Therefore, we decided to use murine organoids to assess the influence of pro-inflammatory stimuli on Dnmt3a gene expression. We have analysed pro-inflammatory marker such as Cxcl1, Cxcl10 and Tnf α (in Suppl. Fig 1c) as confirmation of a successful induction of inflammatory cascades. As a result, we show a specific downregulation of Dnmt3a gene expression upon treatment with TNF α , which is not observed for the other DNA methyltransferase gene Dnmt3b (Suppl. Fig1b). We now clearly state the rationale and limitation of this experiment.

It would be important to describe if DNMT3A AIEC mice given their increased epithelial permeability, show any baseline changes in mucosal immune cell composition and/or phenotype. Information on IEL, mucosal macrophages, and Ig responses would provide important information to the reader. Even if no changes are seen, that is also information that should be added to further understand the implications of DNMT3A function on mucosal homeostasis.

We thank the reviewer for the comment. We performed the suggested analyses on distribution of immune cell types using multispectral FACS in steady state 10 weeks old (n=4) and did not observe significant differences between Dnmt3a fl/fl vs. Dnmt3a Δ IEC mice. This data has been added to the manuscript (Suppl. Fig 3f).

Minor comments:

- In Figure 5c add time point d30?

We have modified the figure as suggested.

- Please provide expression of antimicrobial inflammatory epithelial genes (LCN2, DUOX2, etc).

We thank the reviewer for the comment. We have analyzed the gene expression of the antimicrobial epithelial genes Lcn2, Duox2 and Defb1 in mice from day 5 (acute phase DSS) and day 12 (recovery phase). Gene expression level of Lcn2 and Duox2 are downregulated in Dnmt3a DIEC mice compared to the floxed animals during the recovery phase (suppl. Fig 5e). Same tendency was observed also for Defb1, but it did not reach significance. We hypothesize that during recovery phase lack of induction of protective antimicrobial epithelial genes may contribute to increased tissue damage in the Dnmt3a DIEC mice and added a respective sentence to the discussion section.

- Figure 6a, SEAP activity of serum from non-colitic mice (both WT and KO) should be included or used to show data as FC from baseline

We thank the reviewer for the valid point. We now have analysed SEAP activity from non-colitic mice (steady state) and used the background values to normalize the SEAP activity from mice in the DSS experiment. Results are now depicted as FC (Fig.6d).

- Data in Fig 6b and c would go better in figure 5.

We thank you the reviewer for the suggestion. As we included more data, such as DNAm profiles during acute DSS and immunofluorescence of ZO1 in the chronic colitis experiment, we respectfully decided to rearrange the panels in a different manner to better reflect the sequence of findings.

- Yui et al (ref 44), is not a clinical trial and should not be cited as such. - Also, ref 42 is a trial on teduglutide a GLP-2, not on IL-22

Thanks for pointing out these mistakes, we have replaced the erroneous citations with the appropriate trials.

Reviewer #3 (Remarks to the Author): The study addresses a very interesting and important question in the field of IBD centering on the role of epigenetic regulation of intestinal epithelial cell function. The authors focus on a DNA methylating enzyme (DNMT3A) and use CRISPR/Cas9 targeting of CACO2 cells to generate an in vitro model comparing it to an in vivo model using the Villin-Cre to drive conditional ablation in the murine intestinal epithelium. In analyzing these models, the authors determine the methylation as well as transcriptional network to determine the processes that DNMT3A controls. The manuscript is very well written with very interesting results.

We thank the reviewer for the positive comments.

A criticism of the data is interpretation of DSS susceptibility. Intestinal permeability after DSS treatment increases even in mice without primary defects in tight junctions. Delays in inflammatory clearance could also explain a delay in goblet cell reaccumulation. Overall, there are a number of addressable questions that will improve the quality of the study.

Comments and Suggestions on figures:

- In figure 1, author shows downregulation of DNMT3A in IBD as well as in response to TNF α . Is this specific to DNMT3A or is it seen with other DNA methylases.

The reviewer raises an important point, which refers to the specificity of the observed regulation of DNMT3A. We have analyzed the mRNA levels of the other *de novo* methyltransferase DNMT3B, the regulatory protein DNMT3L and the maintenance methyltransferase DNMT1. Expression levels of DNMT3L were below detection threshold in our sample cohort. We observed no significant regulation of DNMT3B and DNMT1 in the larger new IBD patient cohort compared to the healthy controls, while we confirmed the downregulation of DNMT3A mRNA levels. Additionally, we have analyzed the gene expression levels of *Dnmt3b* and *Dnmt1* in murine organoids upon different inflammatory stimuli and no differences were observed for *Dnmt3b* mRNA levels, while *Dnmt1* was upregulated upon Tnf and LPS stimulation compared to the untreated control.

- Does Figure 1A reflect one or both isoforms of DNMT3A?

The TaqMan assay used for the analysis in Figure 1a and 1b (organoids) can detect both isoforms. In the WB analysis, however, we see a clear prevalence of the DNMT3A1 isoform, which differs by molecular weight. We had pointed out that the observed protein isoform was DNMT3A1 in the previous version, however we now provide uncropped WB images to better depict the differential presence of DNMT3A1 vs. DNMT3A2 in purified primary epithelial cells. We added a sentence to the discussion section that the predominant isoform is DNMT3A1, but that we cannot rule out an additional role for DNMT3A2.

Figure 3:

Authors describe intersection of the CACO2 and in vivo epithelial cell experiments. It would be helpful to see the gene list and degree of expression change perhaps as supplement.

We have now included a table (Supplementary Table 14) with the log fold change comparison

between the significant DEGs *in vitro* and *in vivo*. In addition, we also provide a table with the correlated expression/methylation pairs (mouse vs. human) in Supplementary Table 7.

Figure 5

In figure 5C, are these *Olfm4* and *Ccnd1* tested under DSS treatment?

Yes, the data are derived from chronic DSS treatment. We have clarified this in the legend and the respective panel.

Proliferation is measured. What about cell death?

The question refers to the BrdU staining and the higher proliferative response of the *Dnmt3a* fl/fl vs. *Dnmt3a* Δ IEC mice at day 12 in regenerative phase of the acute DSS experiment. We have performed additional TUNEL staining in colon sections of mice from day 12, and we do not see a significant difference in apoptotic cell count (Suppl.Fig 5a). Specifically, we do not observe increased apoptotic epithelial cells deeper in the crypts, suggesting that the defect in mucosal healing *in vivo* is likely due to reduced proliferation rate rather than increased apoptosis.

Are there differences in permeability to insoluble molecules (e.g., FITC-dextran) of the intestine under steady state conditions *in vivo*? Is this exacerbated under DSS treatment conditions?

This question was raised in a similar manner by reviewer 1. We performed an *in vivo* FITC dextran permeability assay by oral gavage and subsequent measurement of serum concentrations of 10-15 weeks old *Dnmt3a* fl/fl vs. *Dnmt3a* Δ IEC mice under steady state (i.e. untreated) conditions. In this set-up, we now clearly demonstrate that there is increased permeability at baseline in mice with an epithelial deletion of *Dnmt3a* (Fig 4h).

Due to ethical reasons (animal protection law), we are not allowed to repeat a DSS experiment, which already resulted in a clear phenotype (aggravated inflammation, reduced regeneration and higher LPS serum concentration as a proxy for a leaky barrier measured by TLR4-SEAP reporter assay). For the question whether permeability to larger molecules is aggravated under DSS treatment, we thus better describe and discuss our findings on the TLR4 reporter assay, which indicates the sustained presence of higher levels of LPS in the *Dnmt3a* Δ IEC mice at day 12 and therefore hints towards a leaky barrier due to failing regeneration.

The authors show a decrease in some stem cell markers in response to DSS. Is the loss of Goblet cells secondary to a differentiation defect across the secretory lineage? Is there a loss of secretory progenitors?

We thank the reviewer for the comment. We analysed gene expression levels of the secretory progenitor markers *Atoh1* and the proliferative cell marker *Notch1* in *Dnmt3a* Δ IEC mice during acute (day 5) and recovery (day 12) phase. We observed a downregulation of both markers in *Dnmt3a* Δ IEC compared to WT mice during the recovery phase. Notch activation is necessary for mucosal regeneration (Okamoto et al, 2008). Indeed, in WT animals, higher expression of *Notch1* is observed during the recovery phase (day12) while no increased gene expression of *Notch1* is observed in *Dnmt3a* Δ IEC animals. We also see a subtle reduction of Goblet cell numbers already at baseline. These results suggest that diminished Goblet cell numbers might be the consequence of defective secretory cell lineage differentiation, which is partially already present under baseline conditions. We have now included these results in Suppl. Fig5c.

Authors use HEK-BLUE TLR4 assay to measure LPS after DSS treatment. These differences do not appear very large in magnitude. Is there an increase in bacterial translocation to distant sites (e.g., MLN, spleen, liver) after DSS treatment?

We agree that the presentation of the TLR4 reporter assay was suboptimal as it did not include reference values from untreated mice. As this issue was also raised by reviewer #2, we now include the baseline values from untreated mice from the same experimental series and depict the fold change of induction by DSS vs. baseline, which shows a clear difference between Dnmt3a fl/fl vs. Dnmt3a Δ IEC mice at day 12 (recovery phase). We could not directly assess the presence of bacterial translocation after DSS as we were not allowed to repeat the experiment and did not sample the tissues previously. We addressed this point by (a) discussing that the TLR4 serum assay is only a proxy of a leaky barrier and does not directly demonstrate the increased translocation of bacteria in the limitations of study section. (b) We assessed the mucus thickness and presence of 16S rDNA in MLN and spleen in untreated Dnmt3a fl/fl vs. Dnmt3a Δ IEC mice (10-15 weeks of age) and show reduction of mucus thickness as well as increased eubacterial 16S rDNA copies in the MLN already in steady state conditions (Suppl. Fig. 4d and Suppl. Fig. 4e)

Authors measure mRNA levels of key junctional molecules. Are these perturbed at the protein level?

We thank the reviewer for the suggestion. We have now performed immunofluorescence quantification of ZO-1 as a key junctional molecule in colon section of mice during chronic DSS-induced colitis). We observed a reduction of protein level in Dnmt3a Δ IEC mice during chronic DSS colitis. We included these results in Fig.6b.

Authors state that the decrease in PAS positive cells and decreased Muc2 expression would lead to changes in mucous thickness. Is this true in steady state? No direct measure of mucus layer is presented. Does this result in increased bacterial translocation or tissue bound bacteria?

Reply: We again thank the reviewer for valid point, which we addressed by new analyses. (a) We carefully re-analyzed number of PAS positive cells in untreated Dnmt3a fl/fl vs. Dnmt3a Δ IEC mice (10 weeks of age). We demonstrate that a subtle, but significant reduction of Goblet cells is already present at baseline (Suppl. Fig. 4c), arguing for a perturbed differentiation of these secretory cells already under steady state conditions. (b) We performed FISH using a general eubacterial probe (EUB338_Cy3, Microsynth, 3913816) on Carnoy fixed colonic sections from untreated Dnmt3a fl/fl vs. Dnmt3a Δ IEC mice and measured the distance of the luminal bacterial from the epithelial barrier. In line with the reduced numbers of goblet cells, the distance from the epithelium was significantly reduced in Dnmt3a Δ IEC vs. Dnmt3a fl/fl mice, supporting our hypothesis of diminished mucus thickness upon epithelial deletion of Dnmt3a (Suppl. Fig. 4d). We could not unequivocally identify of tissue-bound bacteria (e.g. clear intracellular bacteria in epithelial cells or macrophages) in this staining, regardless of genotype, which we briefly comment on in the respective result section. (c) We performed quantitative real-time PCR using a eubacterial 16S rDNA primer/probe set (Tanner et al., Appl Env Microbiol, 1998) and demonstrate the increased presence of bacterial DNA in MLNs of Dnmt3a Δ IEC mice vs. Dnmt3a fl/fl (Suppl. Fig. 4e). We are aware that this analysis does not directly reflect the translocation of live bacteria, which we noted in the discussion section.

REVIEWERS' COMMENTS

Reviewer #1 (Remarks to the Author):

In this revision, the authors have adequately addressed my previous comments. I recommend for publication at Nature Communication.

Reviewer #2 (Remarks to the Author):

The authors have thoroughly reviewed the manuscript and made significant improvements based on the comments of all 3 reviewers. In particular, the points I raised had been addressed sufficiently. I do not request any further changes.

Reviewer #3 (Remarks to the Author):

Fazio et al provide an interesting and comprehensive analysis of the role of DNMT3A in intestinal epithelial homeostasis and function. The study addresses a very interesting and important question in the field of IBD centering on the role of epigenetic regulation of intestinal epithelial cell function. Since the last submission, the authors have provided additional detail and experiments to strengthen their manuscript. Additional experiments showing changes in barrier function of the mice have been provided (i.e. FITC-Dextran, bacterial translocation to MLN as well as tight junction staining).

Overall, the manuscript is well written and clearly explained. The authors have adequately addressed my initial concerns.

Minor comment:

ZO-1 immunofluorescence imaging of colon is not described in methods. Only general description of IF as used in previous submission of CACO2 cells.